# *Dream2Learn*: STRUCTURED GENERATIVE DREAMING FOR CONTINUAL LEARNING

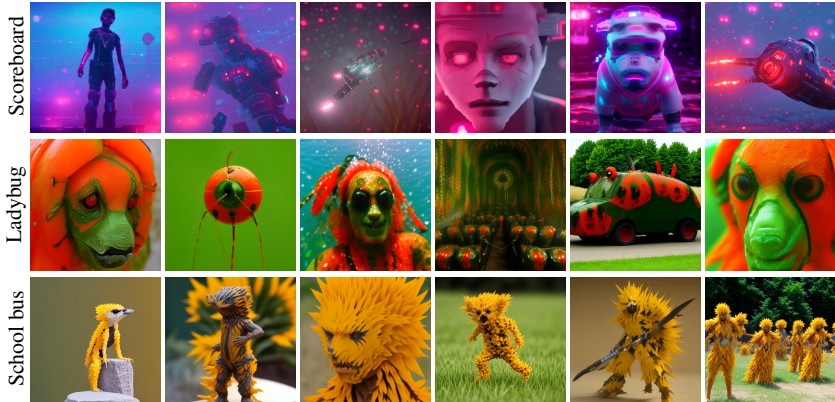

Figure 1: **Dreamed classes generated by D2L**. Examples of dreamed classes synthesized from their corresponding real classes (left). These samples emerge as semantically distinct yet **structurally coherent representations** in the generator's latent space, forming intermediate concepts that enhance the continual classifier's generalization to future tasks.

## ABSTRACT

Continual learning struggles with balancing plasticity and stability while mitigating catastrophic forgetting. Inspired by human sleep and dreaming mechanisms, we propose **Dream2Learn (D2L)**, a generative approach that enables models, trained in a continual learning setting, to synthesize structured additional training signals driven by their internal knowledge. Unlike prior methods that rely on real data to simulate the dreaming process, D2L autonomously constructs semantically distinct yet structurally coherent dreamed classes, conditioning a diffusion model via soft prompt optimization. These dynamically generated samples expand the classifier's representation space, reinforcing past knowledge while structuring features in a way that facilitates adaptation to future tasks. In particular, by integrating dreamed classes into training, D2L enables the model to self-organize its latent space, improving generalization and adaptability to new data. Experiments on Mini-ImageNet, FG-ImageNet, and ImageNet-R show that D2L surpasses existing methods across all evaluated metrics. Notably, it achieves positive forward transfer, confirming its ability to enhance adaptability by structuring representations for future tasks.

## 1 INTRODUCTION

Humans possess a remarkable ability to learn continuously, consolidate past experiences, and generalize knowledge to novel situations (Kumaran et al., 2016; McClelland et al., 1995). This process is also facilitated by memory replay and restructuring during sleep, where the brain synthesizes realistic dreams derived from awake experiences to prepare for future challenges (Ji & Wilson, 2007; Walker & Stickgold, 2004; Singh et al., 2022). In contrast, deep learning models in continual learning (CL) suffer from catastrophic forgetting, wherein previously acquired knowledge deteriorates when new tasks are introduced (McCloskey & Cohen, 1989). Traditional CL methods attempt to address this

issue through rehearsal-based strategies, regularization techniques, or architectural modifications. However, they often struggle to effectively balance stability and plasticity, thereby limiting both long-term knowledge retention and the capacity for adaptation (De Lange & Tuytelaars, 2021; Parisi et al., 2019). Among these, rehearsal-based strategies are widely used due to their ability to stabilize learning by replaying stored examples. Yet, despite their effectiveness, such approaches diverge significantly from how the human brain consolidates memory. Rather than relying on the exact replay of past experiences, the brain engages in generative processes during dreaming, recombining perceptual elements from daily life to construct novel and plausible future scenarios (Llewellyn, 2015; Schwartz, 2003). This process allows for efficient knowledge reinforcement, enabling the brain to improve generalization and anticipate future challenges. Translating this process into artificial neural networks is non-trivial, as it requires the ability to synthesize meaningful and structured representations of the previously learned knowledge without relying on external supervision.

To accomplish this task, in this paper we propose **Dream2Learn (D2L)**, a generative dreaming process that synthesizes training samples directly from the classifier's internal representations. Unlike WSCL (Sorrenti et al., 2024), which relies on surrogate real data to simulate dreams, and other sleep-based approaches that primarily reinforce existing representations (Tadros et al., 2022; Harun et al., 2023), D2L autonomously constructs future-adaptive representations (the *dreams*, indeed), ensuring task relevance and enhancing the model's ability to generalize to new tasks.

As illustrated in Fig. 1, D2L generates structured dreamed classes that serve as intermediate representations, facilitating continual learning. Instead of merely blending past class features, these dreamed samples form coherent yet distinct new concepts, expanding the representation space in a way that supports future task adaptation. By integrating "dreamed classes" into training, the classifier learns high-level reusable features, reinforcing forward transfer while mitigating catastrophic forgetting. This process mirrors the role of REM sleep, where synthetic experiences help refine learned representations, maintaining long-term adaptability as new data distributions emerge.

Our experiments on Mini-ImageNet, FG-ImageNet, and ImageNet-R show that our strategy significantly boosts performance when integrated with standard continual learning methods.

## 2 RELATED WORK

Continual Learning (CL) (De Lange et al., 2021; Parisi et al., 2019) encompasses a family of machine learning techniques that aim to develop models that learn incrementally while avoiding catastrophic forgetting (McCloskey & Cohen, 1989). Common strategies include regularization techniques (Kirkpatrick et al., 2017; Zenke et al., 2017), architectural modifications (Schwarz et al., 2018; Mallya & Lazebnik, 2018), and replay-based methods (Robins, 1995; Rebuffi et al., 2017; Buzzega et al., 2020). More recent approaches enhance model robustness through contrastive learning (Mai et al., 2021; Cha et al., 2021) and latent space regularization (Frascaroli et al., 2024), while experience replay optimizes sample selection for efficient memory retention (Aljundi et al., 2019; Chaudhry et al., 2021). Generative Replay (GR) (Shin et al., 2017; Rios & Itti, 2019; Liu et al., 2020) has emerged as an alternative to buffer-based experience replay by synthesizing past samples, but early methods often faced mode collapse and underperformed compared to traditional replay. Although DDGR (Gao & Liu, 2023), SDDR (Jodelet et al., 2023), and DiffClass (Meng et al., 2024) improve sample fidelity, the role of GR remains retrospective: the generator mainly acts as a memory proxy, reconstructing prior distributions for rehearsal rather than proactively reorganizing representations. Inspired by cognitive neuroscience, several works explore memory mechanisms modeled on brain function. DualNet (Pham et al., 2021) and DualPrompt (Wang et al., 2022a) introduce parallel learning pathways, while CLS-ER (Arani et al., 2022) and FearNet (Kemker & Kanan, 2018) implement short- and long-term memory systems. These approaches focus on stabilizing representations during learning but do not incorporate offline processes for restructuring knowledge. Sleep-based learning offers a complementary perspective, drawing from evidence that wake-sleep cycles refine memory representations (Hinton et al., 1995; Deperrois et al., 2022). Sleep Replay Consolidation (Tadros et al., 2022) applies Hebbian plasticity, and SIESTA (Harun et al., 2023) introduces intermittent consolidation to support online learning. WSCL (Sorrenti et al., 2024) alternates wake and sleep cycles to simulate the benefits of dreaming for memory consolidation. However, instead of generating internal experiences during sleep phases, it shapes the latent space of the classifier using pre-defined representations, limiting the biological plausibility of the dreaming process.

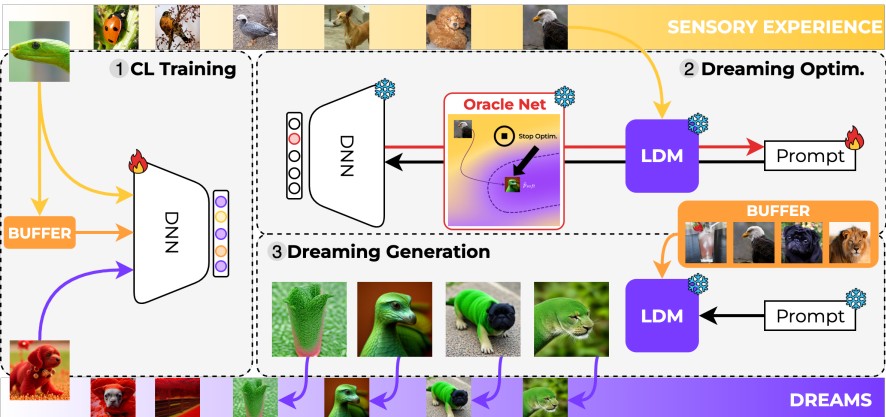

Figure 2: **Overview of Dream2Learn.** (1) During CL training, a deep neural network (DNN) learns from real sensory images (the current task distribution plus the buffer) and dreamed samples produced by a latent diffusion model (LDM). (2) The dreaming optimization process refines the LDM prompts, with an Oracle Network providing a stopping criterion that prevents collapse. (3) Prompts generate auxiliary classes: dreamed samples are not buffered, but rather enrich the representation space with coherent latent clusters that foster knowledge reuse and adaptation (see Appendix A).

D2L reframes generation as a prospective mechanism: rather than replicating past data for rehearsal, generation is used to proactively structure the representation space towards upcoming tasks. It introduces a self-sufficient generative dreaming mechanism, by generating additional training signals from the classifier's internal representations, ensuring task relevance and autonomous dreaming. Through soft prompt optimization, it identifies semantically distinct yet structurally coherent classes in the diffusion model's latent space, which act as intermediate anchors that prime future learning dynamics. Thus, unlike GR techniques that focus on reconstructing or augmenting past distributions (Jodelet et al., 2023; Meng et al., 2024), D2L actively shapes future-adaptive representations, transforming dreaming into a mechanism for fostering forward knowledge transfer and long-term retention, enabling the classifier to adapt more effectively to unseen tasks.

## 3 METHOD

We formulate our continual learning setting as the problem of training a model $F_{\boldsymbol{\theta}}$ over a sequence of $T$ visual classification tasks $\{\tau_1, \ldots, \tau_T\}$, with each task $\tau_t$ associated to a dataset $\mathcal{D}_t = \{(\mathbf{x}_{t,1}, y_{t,1}), \ldots, (\mathbf{x}_{t,n_t}, y_{t,n_t})\}$. Observations $\mathbf{x}_{t,i}$ belong to an image space $\mathcal{I}$, and class sets are disjoint across tasks, i.e., $y_{t,i} \in \mathcal{C}_t$, $\mathcal{C}_j \cap \mathcal{C}_k = \varnothing$. The model's output layer has as many neurons as the total number of classes, i.e., $\sum n_t$. Notation-wise, we will treat $\mathcal{D}_t$ as a probability distribution, when clear from the context.

We also assume the availability of a replay buffer $\mathcal{B}$, where we store a limited number of samples from past tasks for rehearsal, and of a pre-trained and frozen image generator $G$. Our approach requires that $G$ can be conditioned from both textual prompts (with the possibility of adding learnable tokens) and input images; these requirements are easily satisfied by standard text-conditioned latent diffusion models (LDM) — e.g., Stable Diffusion conditioned by CLIP embeddings (Ramesh et al., 2022; Luo et al., 2023) — with an image adapter (Ye et al., 2023)[1]. Formally, $G : \mathcal{I} \times \mathcal{P} \to \mathcal{I}$, with $\mathcal{P}$ being the space of sequences of textual token embeddings[2]. We employ $G$, with appropriate conditioning, to synthesize dream images from past knowledge, thus creating an auxiliary synthetic data stream for preparation to future tasks. At the beginning of our procedure, the model $F_{\boldsymbol{\theta}}$ is trained to learn how to perform task $\tau_1$. Since no knowledge is initially present (as $\boldsymbol{\theta}$ is randomly initialized), we bootstrap the model by training it on task data $\mathcal{D}_1$, optimizing a cross-entropy loss:

$$\min_{\boldsymbol{\theta}} \mathcal{L}_{\text{CE}}(F_{\boldsymbol{\theta}}, \mathcal{D}_1) = -\mathbb{E}_{(\mathbf{x},y) \sim \mathcal{D}_1}\Big[\log p(y|\mathbf{x}; \boldsymbol{\theta})\Big], \tag{1}$$

---

[1]We use h94/IP-Adapter

[2]In practice, $G$ is also made stochastic by receiving a random noise $\epsilon \sim \mathcal{N}(\mathbf{0}, \mathbf{I})$, which is omitted for brevity.

with $p(y|\mathbf{x}; \boldsymbol{\theta})$ being the likelihood of the correct class, given model parameters $\boldsymbol{\theta}$. During this bootstrap phase, we also populate the buffer $\mathcal{B}$ via reservoir sampling (Lopez-Paz & Ranzato, 2017).

The bootstrapped classifier can now be used to synthesize new classes as "dream" variants of what the model has seen up to this point. Dream classes are created by optimizing a learnable prompt $\mathbf{p}_c$ for each class $c \in \mathcal{C}_1$, such that $G(\mathbf{x}, \mathbf{p}_c)$ transforms an input image $\mathbf{x}$ into a "dreamed" versions that vaguely resemble target class $c$, thereby creating a synthetic distribution for a novel "dream class". The details about this optimization process are given in Sec. 3.1. Using this procedure, at task $\tau_1$ we introduce $n_1$ dream classes, i.e., as many as the current task's actual classes. We indicate with $\mathcal{D}_1^d$ the distribution of dream classes defined at this stage.

Let $\mathcal{D}_{\tau_1}$ be the mixture distribution which equally samples from real data $\mathcal{D}_1$ and from the dream distribution $\mathcal{D}_1^d$. The classifier $F_{\boldsymbol{\theta}}$ is then fine-tuned on $\mathcal{D}_{\tau_1}$, replacing $\mathcal{D}_1$ in Eq. 1.

On subsequent tasks $\tau_t$, $t > 1$, we can exploit the model's knowledge on dream classes to ease its learning of new classes. At the beginning of $\tau_t$, we forward samples from each new class $c \in \mathcal{C}_t$ through the model, and map $c$ to the output dream neuron with the largest average likelihood, as in (Bellitto et al., 2022). This allows to bootstrap each class maximizing the reuse of relevant features and preventing disrupting weight updates (details on this procedure in Sec. 3.1). The dream classes corresponding to the assigned classification heads are removed. We then train $F_{\boldsymbol{\theta}}$ on task data $\mathcal{D}_t$ and on $\mathcal{D}_{t-1}^{d*}$, the residual dream distribution obtained from $\mathcal{D}_{t-1}^d$ by removing the discarded dream classes, optimizing the following:

$$\min_{\boldsymbol{\theta}} \Big[ \mathcal{L}_{\text{CE}}(F_{\boldsymbol{\theta}}, \mathcal{D}_{t-1}^{d*} \cup \mathcal{D}_t) + \mathcal{L}_{\text{CL}}(F_{\boldsymbol{\theta}}, \mathcal{D}_t, \mathcal{B}) \Big], \tag{2}$$

where $\mathcal{L}_{\text{CL}}$ is an additional continual learning loss that counters forgetting and explicitly leverages the replay buffer $\mathcal{B}$ for rehearsal. In practice, when sampling from the dream distributions $\mathcal{D}_{t-1}^{d*}$, we employ items stored in the buffer as input conditions to the generator $G$, to increase variability in the dreamed images. Importantly, dreamed samples are never added to $\mathcal{B}$ (refer to Appendix E); only their prompts are retained as part of a persistent dream inventory. After training on task $\tau_t$ and storing rehearsal samples into $\mathcal{B}$, we update the dream inventory for the next task, by optimizing a new set of prompts $\{\mathbf{p}_c \mid c \in \mathcal{C}_t\}$, corresponding to new dream class distributions. The set of $n_t$ newly-generated dream distributions is used to replace an equal number of existing dreaming classes using again the mapping strategy in (Bellitto et al., 2022). We detailed the algorithm in Appendix A.

## 3.1 DREAMING OPTIMIZATION AND MAPPING

The dreaming optimization process for task $\tau_t$ consists of learning a proper conditioning for generator $G$, in order to synthesize samples of novel concepts, expanding the model's representation space while preserving feature reuse.

For each real task class $c \in \mathcal{C}_t$, we aim to generate a corresponding dreamed class $c^d$ that is distinct from all real classes, while contributing to a structured representation in the latent space.

To achieve this, we optimize a learnable prompt $\mathbf{p}_c$ that conditions the generator $G$ to synthesize samples of class $c^d$. Our objective is to identify a transformation trajectory in the LDM latent space such that, given an arbitrary real image $\mathbf{x}$ as input to $G$, the learned prompt $\mathbf{p}_c$ guides $G$ to generate a dreamed version $\mathbf{x}^d$ that shares some characteristics with class $c$, while remaining distinct enough to not be classified as $c$ by the model $F_{\boldsymbol{\theta}}$. This ensures that the dreamed samples populate a structured latent space region that remains visually coherent but semantically separated from real classes.

Formally, we structure the dream class condition $\mathbf{p}_c = [\mathbf{p}_{\text{soft},c}, \mathbf{p}_{\text{text},c}]$ with $\mathbf{p}_{\text{text},c}$ being the fixed text prompt describing the transformation for class $c$, as: "An image of class $[c]$", and $\mathbf{p}_{\text{soft},c}$ being a learnable soft prompt vector optimized to refine the conditioning for class $c$. Due to the stochasticity of $G$, multiple dreamed samples can be generated from the same real image $\mathbf{x}$ and condition $\mathbf{p}_c$. Fig. 3 shows the dreaming optimization process in the LDM latent space.

Prompt optimization is performed by minimizing the cross-entropy loss:

$$\min_{\mathbf{P}_{\text{soft},c}} \mathbb{E}_{\mathbf{x} \sim \mathcal{D}_i \setminus \mathcal{D}_i^c} \Big[ -\log p(c|G(\mathbf{x}, \mathbf{p}_c); \boldsymbol{\theta}) \Big], \tag{3}$$

where $\mathcal{D}_i^c$ is the subset of real samples belonging to class $c$, excluded from the optimization process: this ensures that optimization is not conditioned on images already belonging to the target distribution, allowing the process to gradually converge toward it while generating novel yet structured concepts. As optimization progresses, dreamed samples populate a distinct but structured latent region, allowing future tasks to benefit from enhanced feature reuse and transferability.

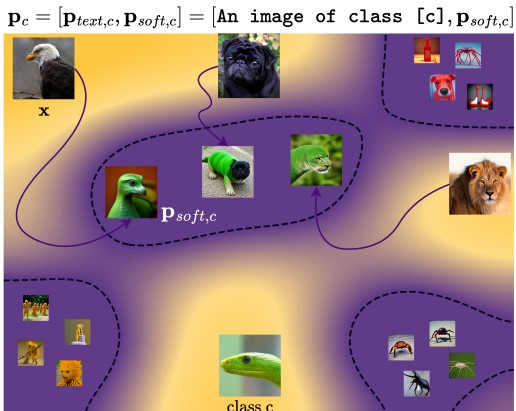

Figure 3: **Visualization of the dreaming optimization process in the latent space of a LDM.**
Given a real sample $\mathbf{x}$, the optimization refines the soft prompt $\mathbf{p}_{\text{soft},c}$ to steer the diffusion model
towards generating a dreamed counterpart that aligns with the target class $c$ (e.g., a green mamba in
this example). The dreaming process explores latent regions where images are visually similar yet
distinct from target classes, forming novel intermediate classes (violet zones).

Dreaming optimization produces a set of conditioning prompts $\{\mathbf{p}_c \mid c \in \mathcal{C}_t\}$ for each class of task $\tau_t$.
Next, we determine the output neurons to which the new dream classes should be mapped, replacing
a subset of dream classes from past tasks. The rationale for this step is to map new classes over
"similar" past dream classes, to ensure a smooth integration and prevent high gradients during training.
Thus, let $\mathcal{C}$ be the set of all output model neurons, and $\mathcal{C}_{\text{real}} = \bigcup_{i \leq t} \mathcal{C}_i$ the set of outputs assigned
to real past tasks. We compute the set of possible destination neurons for the new dream classes as
$\mathcal{C}_{\text{avail}} = \mathcal{C} \setminus \mathcal{C}_{\text{real}}$. Then, we obtain the output neuron $c_{\text{out}}$ for dream class $c^d$ as:

$$c_{\text{out}} = \underset{c \in \mathcal{C}_{\text{avail}}}{\arg \min} \, \mathbb{E}_{\mathbf{x} \sim G_{c^d}} \left[ -\log p(c|\mathbf{x}; \boldsymbol{\theta}) \right], \tag{4}$$

where $G_c$ is the distribution associated to samples produced by $G$ when conditioned with $\mathbf{p}_c$. In
practice, $c^d$ replaces the dream class to which it is more "aligned" in terms of classification likelihood.

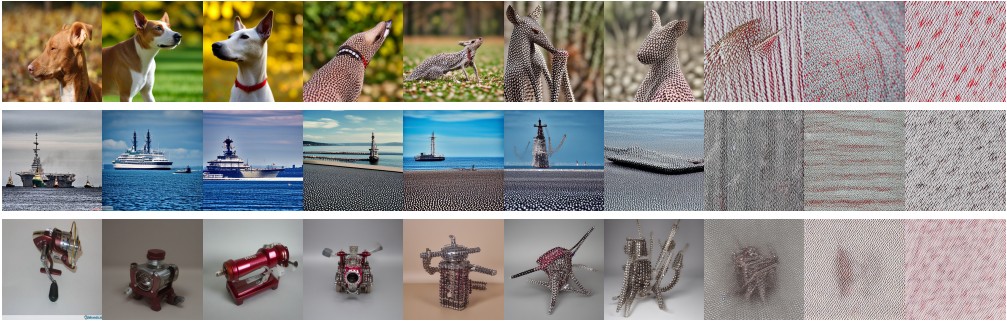

Figure 4: **Examples of dreaming optimization trajectories showing collapse**. From left to right,
the images depict different stages of the optimization process. Each row illustrates the evolution of
three example images throughout the same prompt optimization. Initially, the generated samples
maintain meaningful variations. However, as optimization progresses, they become increasingly
similar, reducing diversity and leading to less effective representations.

## 3.2 ORACLE-GUIDED DREAMING OPTIMIZATION

One key challenge in the dreaming process is determining when to stop optimizing the soft prompt
$\mathbf{p}_{\text{soft},c}$ to avoid collapse or excessive task-specific bias, as illustrated in Fig. 4. If optimization contin-
ues indefinitely toward the convergence of Eq. 3, the generated samples risk becoming redundant or

overfitting to the current task, reducing their effectiveness for future learning. To prevent this, we introduce an oracle network that predicts the optimal stopping point by evaluating whether further refinement of $\mathbf{p}_{\text{soft},c}$ contributes to meaningful latent representation learning. The oracle is trained on a separate dataset $\mathcal{D}_O$, where stopping decisions are labeled based on the quality of generated dreams.

Formally, we define the oracle network $O$, which takes as input a sequence of feature vectors extracted over a temporal window of $k$ optimization steps and provides a binary decision:

$$O(\mathbf{Z}_t) \in \{0, 1\} \tag{5}$$

where $\mathbf{Z}_t$ is the aggregated feature matrix over the last $k$ generated samples:

$$\mathbf{Z}_t = [\mathbf{z}_{t-k+1}, \mathbf{z}_{t-k+2}, \ldots, \mathbf{z}_t] \tag{6}$$

The components of each vector $\mathbf{z}_i \in \mathbb{R}^4$ are the following quantities, computed using the generator $G_c$ conditioned by $\mathbf{p}_{\text{soft},c}$ at the $i$-th optimization iteration of Eq. 3: 1) $\mathbb{E}_{\mathbf{x}}\big[\text{sim}(\mathbf{x}, G_c(\mathbf{x})\big]$, with $\text{sim}(\cdot)$ measuring the structural similarity between the generated image $G_c(\mathbf{x})$ and its conditioning image $\mathbf{x}$, ensuring that the generated image maintains structural coherence; 2) $\mathbb{E}_{\mathbf{x}}\big[f_{\boldsymbol{\theta}}(\mathbf{x})^{\intercal} f_{\boldsymbol{\theta}}\big(G_c(\mathbf{x})\big)\big]$, i.e., the dot product between feature embeddings $f$ extracted by the classifier $F_{\boldsymbol{\theta}}$, capturing the alignment between the representations of $\mathbf{x}$ and $G_c(\mathbf{x})$; 3) $\mathbb{E}_{\mathbf{x}}\big[\mathbf{Q}\big(G_c(\mathbf{x})\big)\big]$, where $Q$ computes the CLIP-based Image Quality Assessment (Wang et al., 2023), evaluating the perceptual quality of the generated image; 4) $\mathbb{E}_{\mathbf{x}}\Big[\sigma\Big(f_{\boldsymbol{\theta}}\big(G_c(\mathbf{x})\big)\Big)\Big]$, i.e., the standard deviation of the feature embeddings, capturing the diversity within generated samples. The optimization process halts once the oracle outputs 1 for $n$ consecutive iterations, ensuring robustness to fluctuations in individual predictions. Once trained, the oracle network $O$ is frozen and used across all tasks to determine when to stop the optimization of $\mathbf{p}_{\text{soft},c}$, ensuring that the dreaming process remains effective without collapsing. Additional training details for the oracle are reported in Appendix B.

## 4 EXPERIMENTAL RESULTS

### 4.1 BENCHMARKS AND TRAINING PROCEDURE

We evaluate D2L on three continual learning benchmarks, obtained by splitting image classification datasets into a series of disjoint tasks:

- **Split Mini-ImageNet** (Vinyals et al., 2016): a widely used few-shot learning dataset, consisting of ImageNet 100 classes with 600 samples each, commonly adapted for continual learning;
- **Split FG-ImageNet** (Russakovsky et al., 2015)[3], a fine-grained image classification benchmark with 100 animal classes from ImageNet, designed to evaluate continual learning methods on a more challenging task.
- **Split ImageNet-R** (Hendrycks et al., 2021) comprises various renditions of 200 ImageNet classes (e.g., paintings, sculptures, embroidery, cartoons, etc.), with 150 samples each, introducing strong intra-class variations.

In our experimental setup, half of the classes in each dataset are used for the first task, while the remaining classes are equally split across the subsequent tasks. In particular, excluding the first task, the Mini-ImageNet and FG-ImageNet datasets consist of 10 tasks with 5 classes each, whereas the ImageNet-R dataset consists of 5 tasks with 20 classes each.

In terms of training procedure, we adopt ResNet-18 (He et al., 2016) as the backbone and train for 10 epochs per task using SGD (learning rate 0.03, batch size 32). Given the large number of classes, we use buffer sizes of 2000 and 5000.

Prompt optimization is performed in Stable Diffusion's text space via cross-entropy loss, guided by classifier predictions. We use Adam (learning rate 0.1, batch size 1), with the number of iterations controlled by an oracle network. The oracle is a single-hidden-layer MLP trained on a labeled dream quality dataset $\mathcal{D}_O$ (see Sec. 3.2) using Adam (0.001, 500 epochs). It stops optimization when a termination signal is predicted in at least n = 2 of the past k = 3 iterations. We use ImageNet-R as $\mathcal{D}_O$ when testing on Mini-ImageNet or FG-ImageNet, and FG-ImageNet for ImageNet-R. Results are for class-incremental setting, reported as mean ± std over 5 runs.

---

[3]Split FG-ImageNet is derived from `https://www.kaggle.com/datasets/ambityga/imagenet100`

## 4.2 RESULTS

To assess the impact of our method, we evaluate its effectiveness when applied in conjunction to continual learning state-of-the-art models. Since the dream generation mechanism relies on combining learned prompts with past experiences stored in the buffer, we apply it exclusively in conjunction with rehearsal-based methods. Specifically, we consider DER++ (Buzzega et al., 2020), ER-ACE (Caccia et al., 2022), and ER (Chaudhry et al., 2019b), comparing their performance with and without the dreaming generation. Tab. 1 presents results in terms of *final average accuracy (FAA)* in the class-incremental setting, i.e., the accuracy on a separate test set including all task classes, measured after training on the last task, with no knowledge on task identity at inference time. Our approach leads to a significant improvement in performance across all benchmarks, demonstrating the importance of mimicking human dreaming for mitigating forgetting.

Table 1: **Class-incremental final average accuracy (FAA) of rehearsal-based methods**, with and without dreaming, for buffer sizes 2000 and 5000.

|  | Mini-ImageNet | | FG-ImageNet | | ImageNet-R | |
|---|---|---|---|---|---|---|
|  | 2000 | 5000 | 2000 | 5000 | 2000 | 5000 |
| ER | 27.91±3.49 | 34.21±3.04 | 21.08±2.38 | 22.21±3.44 | 7.68±0.97 | 10.69±1.29 |
| ↪D2L | **31.18**±2.74 | **39.75**±2.61 | **23.53**±1.98 | **32.73**±3.39 | **8.67**±0.66 | **11.84**±0.95 |
| DER++ | 14.74±2.14 | 26.92±4.72 | 14.43±3.68 | 23.86±2.54 | 6.08±0.81 | 8.29±1.15 |
| ↪D2L | **21.06**±5.45 | **31.91**±5.19 | **18.86**±3.22 | **25.38**±2.17 | **8.60**±1.00 | **10.89**±1.56 |
| ER-ACE | 33.26±3.51 | 40.59±1.20 | 24.79±5.02 | 30.16±4.97 | 7.09±0.59 | 9.44±0.70 |
| ↪D2L | **40.90**±0.95 | **47.32**±0.89 | **31.57**±1.20 | **38.50**±1.01 | **9.54**±0.39 | **12.51**±0.56 |

Table 2: **Forward Transfer (FWT) of rehearsal-based methods**, with and without dreaming, for buffer sizes 2000 and 5000.

|  | Mini-ImageNet | | FG-ImageNet | | ImageNet-R | |
|---|---|---|---|---|---|---|
|  | 2000 | 5000 | 2000 | 5000 | 2000 | 5000 |
| ER | -2.58 | -1.62 | -1.88 | -1.52 | -1.32 | -0.64 |
| ↪D2L | **+0.33** | **+0.47** | **+1.79** | **+1.19** | **+0.24** | **+0.14** |
| DER++ | -1.55 | -2.00 | -1.36 | -2.48 | -1.03 | -1.83 |
| ↪D2L | **+0.97** | **+0.71** | **-0.13** | **+1.86** | **+0.30** | **+0.24** |
| ER-ACE | -1.99 | -2.45 | -2.00 | -2.16 | -2.46 | -1.27 |
| ↪D2L | **+1.05** | **-1.58** | **+1.09** | **+0.08** | **-0.25** | **+0.17** |

One of our key claims is that our dreaming mechanism enhances a model's ability to prepare for future tasks. To validate this, we evaluate forward transfer (FWT) (Lopez-Paz & Ranzato, 2017), measuring how well the model leverages prior knowledge when learning new tasks. FWT is defined as the average difference between the accuracy on a task $\tau_t$ by a model trained up to $\tau_{t-1}$, and the accuracy on $\tau_t$ by a randomly initialized model. Since a continually trained model often predicts known classes, FWT is typically negative. Table 2 shows that dream generation improves FWT, with D2L achieving positive forward transfer in some cases, similar to WSCL. However, WSCL achieves positive forward transfer by relying on additional real data to simulate dreams, whereas D2L internally generates these dreams by leveraging the model's own internal knowledge. Furthermore, we conduct a comprehensive performance analysis by comparing the best performing version of our approach from Tab. 1 (i.e., ER-ACE + D2L) with state-of-the-art continual learning methods: [4] GSS (Aljundi et al., 2019), A-GEM (Chaudhry et al., 2019a), iCaRL (Rebuffi et al., 2017), FDR (Benjamin et al., 2019), BiC (Wu et al., 2019), and RPC (Pernici et al., 2021). Results are shown in Table 3. To contextualize these results, we also define a lower bound as training without any countermeasure to forgetting (*Fine-tune*). ER-ACE + D2L outperforms state-of-the-art methods across all examined datasets and

---

[4]Results were obtained using the original code released alongside the corresponding papers.

Table 3: **Comparison with state-of-the-art methods, in terms of class-incremental final average accuracy (FAA)**, for buffer sizes 2000 and 5000.

| Method | Mini-ImageNet | | FG-ImageNet | | ImageNet-R | |
|---|---|---|---|---|---|---|
| Fine-tune | 6.72±1.20 | | 6.98±0.10 | | 4.46±0.15 | |
| **Buffer-based methods** | | | | | | |
| | **2000** | **5000** | **2000** | **5000** | **2000** | **5000** |
| GSS | 6.40±0.38 | 5.71±0.08 | 8.07±0.26 | 9.23±0.85 | 5.08±0.13 | 4.29±0.32 |
| A-GEM | 6.78±1.13 | 7.45±0.76 | 6.20±1.11 | 6.11±1.13 | 4.69±0.03 | 6.29±0.84 |
| RPC | 9.22±0.30 | 9.02±0.24 | 8.13±0.11 | 7.41±0.74 | 5.71±0.03 | 6.32±0.80 |
| DER++ | 14.74±2.14 | 26.92±4.72 | 14.43±3.68 | 23.86±2.54 | 6.08±0.81 | 8.29±1.15 |
| FDR | 15.46±1.10 | 11.58±0.96 | 9.17±2.40 | 12.91±0.95 | 5.71±0.18 | 5.77±0.10 |
| iCaRL | 16.46±0.51 | 16.50±0.33 | 8.54±0.88 | 8.86±0.25 | 1.97±0.28 | 1.91±0.29 |
| ER | 27.91±3.49 | 34.21±3.04 | 21.08±2.38 | 22.21±3.44 | 7.68±0.97 | 10.69±1.29 |
| BiC | 30.56±7.41 | 37.84±0.61 | 27.83±2.75 | 32.29±0.70 | 7.15±1.14 | 8.60±2.07 |
| ER-ACE | 33.26±3.51 | 40.59±1.20 | 24.79±5.02 | 30.16±4.97 | 7.09±0.59 | 9.44±0.70 |
| **ER-ACE + D2L** | **40.90**±0.95 | **47.32**±0.89 | **31.57**±1.20 | **38.50**±1.01 | **9.54**±0.39 | **12.51**±0.56 |

Table 4: **Ablation on the oracle.** Results on Mini-ImageNet comparing our method with Fixed optimization.

| Method | Buffer size | |
|---|---|---|
| | **2000** | **5000** |
| ER-ACE baseline | 33.26±3.51 | 40.59±1.20 |
| + Fixed optim. | 38.94±0.97 | 46.41±0.55 |
| + **Oracle (D2L)** | 40.90±0.95 | 47.32±0.89 |

Table 5: **Impact of dream class updates.** Comparison on Mini-ImageNet of different dreaming strategies.

| Dreaming | Buffer size | |
|---|---|---|
| | **2000** | **5000** |
| No dreams | 33.36±3.51 | 40.59±1.20 |
| At beginning | 36.93±2.09 | 41.59±2.54 |
| Incremental | 36.85±1.16 | 43.87±2.90 |
| **D2L** | **40.90**±0.95 | **47.32**±0.89 |

buffer sizes, with significant margins. Note that our method targets continual learning from scratch with a randomly initialized convolutional backbone, without external pre-training. By contrast, a separate line of work relies on pre-trained models, either via full/partial fine-tuning (Ramasesh et al., 2021; Boschini et al., 2022; Ostapenko et al., 2022) or prompt tuning (Wang et al., 2022b; Smith et al., 2023) that adapts frozen ViT backbones. These approaches operate in a different evaluation regime, adapting an already rich representation and thus conflating the effect of the CL strategy with benefits from large-scale pre-training and transformer inductive biases. In our setting, the classifier's emergent knowledge directly guides the dreaming generation process, so mixing regimes would not yield an informative head-to-head comparison. For clarity and fairness, we therefore restrict comparison to methods that, like ours, train a CNN backbone from scratch under the same protocol constraints.

### 4.3 MODEL ANALYSIS

Model analysis is primarily conducted to assess the contribution of the dream generation strategy. The ER-ACE model (Chaudhry et al., 2019b), identified as the top-performing method when combined with our approach (see Tab. 1), is used as the baseline model for this study. All experiments are performed on the Mini-ImageNet dataset (Vinyals et al., 2016).

We first ablate the oracle and consider *Fixed optimization*, where prompt updates stop once the classifier predicts the target class for four consecutive steps. This criterion assumes the trajectory has reached and stabilized in the target representation space, but it limits adaptability; our full method avoids this issue and achieves superior performance (Tab. 4). Then, we assess how updating dreams during the sequential learning of tasks affects the overall performance. Our default strategy creates new dream classes at the end of each task—equal in number to that task's classes—and replaces

an equal number of old ones, keeping the classifier output size fixed. We compare this with three variants: (1) *No dreams*, a baseline without generated classes; (2) *At beginning only*, where dream classes are generated once during the first task and reused throughout; and (3) *Incremental*, which accumulates dream classes across tasks, expanding the output layer with weights sampled from $\mathcal{N}(\mu_{\mathbf{w}}; \sigma_{\mathbf{w}}^2)$ based on existing neurons. As shown in Tab. 5, our replacement strategy yields the best results. Fixed dreams limit forward transfer by excluding later tasks, while incremental expansion likely degrades performance due to increased task complexity.

We also compare D2L with WSCL (Sorrenti et al., 2024) on Mini-ImageNet under different buffer sizes. While WSCL achieves slightly higher accuracy (compare to Tab. 4): 42.38% (±1.16) with a 2000-sized buffer and 48.30% (±2.60) with a 5000-sized buffer, it does so by using an auxiliary dataset to pre-activate future task heads. This mechanism, while effective, does not simulate a true dreaming process. In contrast, D2L explicitly models dreaming by generating task-relevant samples guided by the classifier's internal knowledge—mirroring how humans rely on memory-driven simulations to reinforce learning.

To further validate our method, we also replaced the dreaming mechanism with alternative strategies for creating class blends, namely Mixup-based methods applied either in image space or in latent/text diffusion space. The corresponding results are reported in the Appendix C. Finally, we want to support that dreams generated by D2L come from the classifier's internal knowledge and cannot unintentionally resemble future classes. To this aim we assess if dreams are truly out-of-distribution (OOD) with respect to the target dataset. In particular, we generate dreams using our method on the Mini-ImageNet benchmark and classify each generated image using a ViT-B/16 model pre-trained on ImageNet-1K. This model acts as an external expert not involved in the training process, mapping each dream to one of the 1000 ImageNet classes. We then measure how many dreams are classified into the 100 classes used in Mini-ImageNet, and how many are mapped to the remaining 900 classes. Results show that only 10.11% of the dreams are assigned to the 100 classes in Mini-ImageNet, while 89.89% are mapped to the other 900 classes. This roughly matches the underlying class prior (10% vs. 90%), suggesting that dreams are OOD relative to Mini-ImageNet images and therefore cannot qualitatively anticipate the target classes.

## 5 LIMITATIONS

While D2L–generated dreams are qualitatively distant from the target dataset, a residual limitation is the potential categorical leakage of future-class information. To quantify this, we performed a *leak test* using a joint classifier trained on all classes in the sequence. A *leak* is counted when a future class is assigned to a head previously seeded by a dream that the joint model also maps to that class. On MiniImageNet experiments, we observe 1.76 *leaks* on average (3.93% of class replacements). We remark that these events are not only rare but also contingent on the premise that Stable Diffusion can faithfully synthesise future classes—a premise the OOD-distribution experiment fails to support. Nonethenless, mitigation of this theoritical leakage could involve unlearning future-class concepts from the diffusion model—e.g., through concept-erasure or negative-gradient editing—prior to prompt optimization.

## 6 CONCLUSION

In this work, we introduced **Dream2Learn**, an approach inspired by the ability of the human brain to consolidate past experiences and anticipate future ones through dreaming. Our method pairs a classification network with a generative model to synthesize structured training signals, reinforcing past knowledge and enhancing forward transfer—allowing the model to leverage prior knowledge to improve learning on future tasks. Experiments on standard continual learning benchmarks show that dreaming helps mitigate forgetting and can support feature learning by expanding the classifier's representation space, turning negative forward transfer into positive. Using soft prompt optimization within a latent diffusion model, D2L generates novel yet coherent classes with the oracle model helping to maintain sample quality by preventing collapse.

In summary, D2L offers a practical approach to structuring model knowledge over time. By generating intermediate representations, it illustrates the potential of synthetic data to support abstraction and transfer in continual learning.

**Reproducibility Statement.** All CL experiments are implemented on top of the Mammoth framework `https://github.com/aimagelab/mammoth`; baseline results are obtained using the implementations provided within the framework under the same memory and protocol constraints. Data pre-processing steps, data splits, used hyperparameters mirroring our setup are documentd in the Appendix F. Assumptions, architectural choices and training details for the oracle network are explained in the main text and expanded in the Appendix B. The full codebase for prompt optimization, generation and experiment scripts will be publicly released upon acceptance.

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

# APPENDICES

## A  METHOD ALGORITHM

Algorithm 1 delineates the D2L pipeline introduced in Sec. 3. For clarity of presentation, the pipeline is simplified by omitting the initial and final tasks. During the initial task, the absence of dreaming classes reduces the training loss to Eq. 1, while the final task does not perform dreaming generation and optimization as these phases are unnecessary.

---

**Algorithm 1** Dream2Learn (D2L)

---

**Notation**

    $T$, the number of task

    $\mathcal{C}_t$, the classes of task $t$

    $F_{\boldsymbol{\theta}}$, the continual classifier

    $G$, the generator

    $\mathcal{B}$, the buffer

    $\mathcal{D}_t$, the real data distribution at task $t$

    $\mathcal{D}_{t-1}^d$ the *dream* distribution used during continual training at task $t$

    $\mathcal{D}_t^r$, the distribution of the dream classes to be *removed* at task $t$

    $\mathcal{D}_{t-1}^{d*}$, the residual *dream* classes after mapping at task $t$

    $\mathbf{x}$, a real image

    $\mathbf{x}^d$, a generated *dream* image

    $\mathbf{p}_c$, the learnable prompt associated with class $c$

    $\mathcal{D}_{t,c}^d$, the distribution of *dreams* generated after task $t$ from class $c$

    $\mathcal{D}_{t,\mathcal{C}_t}^d$, the distribution of *dreams* generated after task $t$ from classes $\mathcal{C}_t$

    $\mathcal{D}_t^d$, the distribution of all *dreams* after task $t$

---

1: **for** $t = 2$ **to** $T - 1$ **do**
2:     $\mathcal{D}_t^r \leftarrow Mapping(F_{\boldsymbol{\theta}}, \mathcal{D}_t)$         ▷ *real classes mapping*
3:     $\mathcal{D}_{t-1}^{d*} \leftarrow \mathcal{D}_{t-1}^d \setminus \mathcal{D}_t^r$
4:     **for all** epochs **do**         ▷ *CL training*
5:         $loss \leftarrow \mathcal{L}_{CE}(F_{\boldsymbol{\theta}}, \mathcal{D}_{t-1}^{d*} \cup \mathcal{D}_t) + \mathcal{L}_{\text{CL}}(F_{\boldsymbol{\theta}}, \mathcal{D}_t, \mathcal{B})$
6:         $\mathcal{B} \leftarrow ReservoirSample(\mathcal{B}, \mathcal{D}_t)$
7:         **update** $\theta$
8:     **end for**
9:     **for all** $c \in \mathcal{C}_t$ **do**         ▷ *dreaming optimization*
10:         **repeat**
11:             $\mathbf{x}^d \leftarrow G(\mathbf{x}, \mathbf{p}_c)$
12:             $loss \leftarrow \mathcal{L}_{CE}(F_{\boldsymbol{\theta}}, (\mathbf{x}^d, c))$
13:             **update** $\mathbf{p}_c$
14:             $stop \leftarrow Oracle(\mathbf{x}, \mathbf{x}^d)$
15:         **until** $stop$
16:         $\mathcal{D}_{t,c}^d \leftarrow Generate(G, \mathcal{B}, \mathbf{p}_c)$         ▷ *dreaming generation*
17:     **end for**
18:     $\mathcal{D}_{t,\mathcal{C}_t}^d \leftarrow \bigcup_c \mathcal{D}_{t,c}^d$
19:     $\mathcal{D}_t^r \leftarrow Mapping(F_{\boldsymbol{\theta}}, \mathcal{D}_{t,\mathcal{C}_t}^d)$         ▷ *dream classes mapping*
20:     $\mathcal{D}_t^d \leftarrow (\mathcal{D}_{t-1}^{d*} \setminus \mathcal{D}_t^r) \cup \mathcal{D}_{t,\mathcal{C}_t}^d$
21: **end for**

---

## B  ORACLE TRAINING AND GENERALIZATION

In the main paper (Sec. 3.2) we introduced the oracle network $O$, whose goal is to determine the stopping point during prompt optimization. The formal definition of $O$ is already provided in the main manuscript; here we detail its training procedure.

## B.1  DATASETS

To avoid semantic overlap between the oracle's training data and the continual evaluation benchmarks, we always trained the oracle on classes from a disjoint dataset. Concretely:

- When the benchmark was Mini-ImageNet or FG-ImageNet, the oracle was trained on dreaming trajectories from ImageNet-R;
- When the benchmark was ImageNet-R, the oracle was trained on dreaming trajectories from FG-ImageNet.

This ensures that the oracle never sees classes related to the benchmarks in which it is applied, thus preventing task-specific bias.

Each trajectory was obtained by optimizing the soft prompt $\mathbf{p}_{\text{soft},c}$ for up to 500 steps. The stopping point was labeled as the iteration at which generated samples exhibited both high perceptual quality and sufficient diversity, while avoiding collapse or excessive specialization.

## B.2  FEATURES

The oracle network takes as input feature sequences $\mathbf{Z}_t = [\mathbf{z}_{t-k+1}, \ldots, \mathbf{z}_t]$, with each $\mathbf{z}_i \in \mathbb{R}^4$ summarizing properties of the generated samples at iteration $i$. In order to train the oracle network $O$, we initially designed a pool of 25 candidate features capturing different aspects of the generation process. These features can be grouped into three broad categories:

- **Image-level quality metrics**. We compute average SSIM, PSNR, and MSE among pairs of generated images at the same optimization step. These same metrics are also computed between each generated image and its conditioning (target) image. We also compute CLIP-iQA scores (quality, complexity, naturalness, realism) on both generated and target images.
- **Feature-based statistics**. We extract classifier feature-level statistics: cosine similarity and MSE computed either among generated images or between generated and conditioning images. We therefore computed embeddings standard deviation on both generated and target images.
- **Classifier-based uncertainty signals.** From the target classifier logits we compute statistical descriptors including variance, entropy, range (max–min), and kurtosis, averaged across generated samples. We also include the cross-entropy loss signal used during prompt optimization.

To reduce redundancy and identify the most informative subset, we performed a SHAP-based feature importance analysis across multiple trajectories. Four features consistently ranked highest and were retained for the oracle used in the main experiments:

1. SSIM between conditioning and generated images (structural fidelity).
2. Cosine similarity between classifier embeddings of conditioning and generated images (semantic alignment).
3. CLIP-iQA quality score of generated images (perceptual quality).
4. Standard deviation of classifier embeddings within generated samples (diversity).

These four features jointly capture complementary aspects of generation dynamics: (i) structural coherence, (ii) semantic alignment, (iii) perceptual quality, and (iv) diversity. Importantly, they are not tied to dataset-specific semantics, which explains why the oracle generalizes robustly across benchmarks even when trained only once on disjoint classes.

## B.3  MODEL

The oracle $O$ is implemented as a lightweight multilayer perceptron (MLP). It receives as input the temporal feature sequence $\mathbf{Z}_t = [\mathbf{z}_{t-k+1}, \ldots, \mathbf{z}_t]$, where each $\mathbf{z}_i \in \mathbb{R}^4$ contains the metrics described in Sec. B.2. The network consists of a single hidden layer with 32 units, ReLU activation, and a sigmoid output for binary classification.

Table 6: **Ablation on dream generation strategies.** Evaluation on Mini-ImageNet comparing interpolation-based baselines with our proposed D2L.

| Method | Buffer size | |
|---|---|---|
| | 2000 | 5000 |
| ER-ACE baseline | 33.26±3.51 | 40.59±1.20 |
| + Mixup | 36.84±0.94 | 44.82±1.27 |
| + Continual Mixup | 36.05±1.22 | 43.45±0.83 |
| + Textual Mixup | 31.89±0.50 | – |
| + Synth Mixup | 36.99±0.26 | – |
| + **Oracle (D2L)** | **40.90**±0.95 | **47.32**±0.89 |

The model is trained supervisedly on labeled dreaming trajectories with the Adam optimizer for a maximum of 500 iterations, using binary cross-entropy loss. A validation split is employed for early stopping. Once trained, the oracle is frozen and used across a sequence of tasks including only classes never seen during its training. Its inference overhead is negligible (less than 1 ms per step), making it effectively free compared to prompt optimization and dream generation.

### B.4 GENERALIZATION

To assess the generalization capability of the oracle, we trained a single model on a dedicated dream-quality dataset constructed from 100 ImageNet classes, disjoint from all benchmarks used in the main experiments (Mini-ImageNet, FG-ImageNet, and ImageNet-R). Labels were assigned by inspecting dreaming trajectories and selecting the iteration that yielded high-quality and diverse images without collapse.

This general-purpose oracle was then applied across all tasks and datasets, without retraining or adaptation. The predicted stopping points differed from those obtained with dataset-specific oracles by only 9.28 iterations on average (out of 500 optimization steps). Importantly, no trajectory collapse was observed and continual learning performance remained unchanged.

These results confirm that oracle training does not need to be repeated for each benchmark: a single instance trained once on a disjoint dataset generalizes robustly. This is explained by the choice of features—structural similarity, embedding alignment, CLIP-based perceptual quality, and embedding diversity—which capture dataset-agnostic properties of generation dynamics.

## C COMPARISON WITH MIXUP-BASED STRATEGIES

In principle, the dreaming process could be replaced by surrogate samples generated from past knowledge using interpolation-based techniques. To test this hypothesis, we substituted our dreamed classes with synthetic ones obtained through different Mixup strategies and related baselines.

- **Mixup** (Zhang et al., 2018): combines images from the current data stream with samples from the replay buffer to form auxiliary synthetic classes.
- **Continual Mixup**: applies the same interpolation scheme, but only between images sampled from the replay buffer.
- **Textual Mixup**: interpolates text embeddings of class prompts, producing mixed conditioning signals without structural generation.
- **Synth Mixup**: interpolates directly in the latent space of the diffusion model.

As shown in Table 6, D2L consistently outperforms all interpolation-based strategies. In contrast to Mixup variants, which rely on static blending of existing representations, D2L produces distinct and task-aware latent clusters. Importantly, our generation process is explicitly guided by the knowledge encoded in the classifier.

## D    EVALUATION ACROSS MULTIPLE TASK ORDERINGS

While earlier experiments considered a fixed task sequence, we also evaluate D2L under multiple random class orderings to obtain a more robust assessment of its generalization across datasets. Specifically, we repeated the main experiments using three different class orderings (random seeds: 1607, 23, and 0) with a buffer size of 2000. We report mean and standard deviation across these runs. Results in Table 7 confirm that D2L consistently outperforms the baseline across all datasets, showing that its benefits are stable and not tied to a particular task sequence.

Table 7: **Evaluation under multiple task orderings.** Results are averaged over three random class sequences (seeds: 1607, 23, 0) with buffer size 2000. D2L consistently outperforms the baseline.

| Dataset | Baseline (mean $\pm$ std) | D2L (mean $\pm$ std) |
|---|---|---|
| SeqMINIIMG | $32.39 \pm 1.25$ | $\mathbf{39.42 \pm 2.42}$ |
| SeqImageNet-FG | $27.18 \pm 2.08$ | $\mathbf{33.32 \pm 1.54}$ |
| SeqImageNet-R | $8.28 \pm 1.03$ | $\mathbf{9.93 \pm 0.41}$ |

## E    GENERATIVE REPLAY VS. DREAM2LEARN

In this section, we aim to clarify the conceptual differences between *generative replay* (GR) methods and our proposed Dream2Learn (D2L), and we provide additional experimental results for completeness.

GR methods such as DGR (Shin et al., 2017) or DDGR (Yan et al., 2023) discard the memory buffer and rely entirely on a generative model to reconstruct past data, with the objective of preserving knowledge of previous tasks through explicit replay. In contrast, D2L differs along two fundamental dimensions:

- **Buffer.** Unlike GR, D2L retains a fixed-size buffer (e.g., 2000 samples), ensuring direct access to real exemplars throughout training. Nonetheless, **buffer never contains generated images**.

- **Generation.** While GR employs generation to reproduce past samples that directly replace the buffer, D2L leverages generation in a profoundly different way: generated images in D2L do not serve as memory replacements, but as additional data stream (additional classes) that pre-activate future-class representations.

This conceptual divergence means that direct comparisons should be interpreted carefully, as the underlying objectives and mechanisms are not the same. Nevertheless, since both approaches involve generative components during training and thus incur comparable overheads, we report results against representative GR methods under the same buffer constraint, as shown in Tab. 8.

Table 8: Comparison with generative replay methods under the same buffer constraint (2000 samples). While D2L is not a generative replay method, its use of generation for anticipatory transfer leads to superior performance compared to GR approaches.

| Method | SeqMiniImageNet | SeqImageNet-FG | SeqImageNet-R |
|---|---|---|---|
| DGR (Shin et al., 2017) | $23.33 \pm 0.32$ | $26.17 \pm 0.21$ | $7.00 \pm 0.26$ |
| DDGR (Yan et al., 2023) | $37.48 \pm 0.98$ | $30.81 \pm 0.54$ | $9.21 \pm 0.17$ |
| **ER-ACE + D2L (ours)** | $\mathbf{40.90 \pm 0.95}$ | $\mathbf{31.57 \pm 1.20}$ | $\mathbf{9.54 \pm 0.39}$ |

These results confirm that D2L achieves higher accuracy than GR approaches, despite pursuing a different goal. Whereas GR attempts to reconstruct and replay the past, D2L leverages anticipatory generation to expand and stabilize the representation space, proving more effective across all benchmarks.

# F    REPRODUCIBILITY DETAILS

## F.1    ADDITIONAL TRAINING DETAILS

All experiments are conducted on a workstation with 384-core CPU, 1TB RAM and 4 NVIDIA H100 GPUs. The continual learning methods are trained on a single NVIDIA H100 GPU. As a reference, each training run with ER-ACE on Mini-ImageNet requires approximately 3 hours. The dream generation process, based on the use of Stable Diffusion, is the most computationally intensive part. To this aim, we use 4 NVIDIA H100 GPUs and it requires approximately 6 hours per run (including both prompt optimization and dreams generation), with PyTorch DistributedDataParallel. Results in Sec. 4.2 are reported in terms of mean and standard deviation over five runs with different random seeds.

## F.2    HYPERPARAMETER SEARCH

In Tables 9, 10 and 11 we show the best hyperparameters combinations for each method.

Table 9: Mini-ImageNet

| Method | Buffer | **Mini-ImageNet** |
|---|---|---|
| Fine-tune | – | lr: 0.03 |
| ER | 2000 | lr: 0.03; |
| ER | 5000 | lr: 0.03; |
| DER++ | 2000 | lr: 0.01; alpha: 0.1; beta: 0.5; |
| DER++ | 5000 | lr: 0.01; alpha: 0.1; beta: 0.5; |
| ER-ACE | 2000 | lr: 0.01; mom: 0 wd: 0 |
| ER-ACE | 5000 | lr: 0.01; mom: 0 wd: 0 |
| GSS | 2000 | lr: 0.03; |
| GSS | 5000 | lr: 0.03; |
| A-GEM | 2000 | lr: 0.03; |
| A-GEM | 5000 | lr: 0.03; |
| RPC | 2000 | lr: 0.03; |
| RPC | 5000 | lr: 0.03; |
| FDR | 2000 | lr: 0.03; alpha:0.3; |
| FDR | 5000 | lr: 0.03; alpha:0.3; |
| iCaRL | 2000 | lr: 0.03; |
| iCaRL | 5000 | lr: 0.03; |
| BiC | 2000 | lr: 0.03; |
| BiC | 5000 | lr: 0.03; |

Table 10: FG-ImageNet

| Method | Buffer | **FG-ImageNet** |
|---|---|---|
| Fine-tune | – | lr: 0.03 |
| ER | 2000 | lr: 0.03; |
| ER | 5000 | lr: 0.03; |
| DER++ | 2000 | lr: 0.03; alpha: 0.1; beta: 0.5; |
| DER++ | 5000 | lr: 0.03; alpha: 0.1; beta: 0.5; |
| ER-ACE | 2000 | lr: 0.03; mom: 0 wd: 0 |
| ER-ACE | 5000 | lr: 0.03; mom: 0 wd: 0 |
| GSS | 2000 | lr: 0.03; |
| GSS | 5000 | lr: 0.03; |
| A-GEM | 2000 | lr: 0.03; |
| A-GEM | 5000 | lr: 0.03; |
| RPC | 2000 | lr: 0.03; |
| RPC | 5000 | lr: 0.03; |
| FDR | 2000 | lr: 0.03; alpha:0.3; |
| FDR | 5000 | lr: 0.03; alpha:0.3; |
| iCaRL | 2000 | lr: 0.03; |
| iCaRL | 5000 | lr: 0.03; |
| BiC | 2000 | lr: 0.03; |
| BiC | 5000 | lr: 0.03; |

Table 11: ImageNet-R

| Method | Buffer | **ImageNet-R** |
|---|---|---|
| Fine-tune | – | lr: 0.03 |
| ER | 2000 | lr: 0.03; |
| ER | 5000 | lr: 0.03; |
| DER++ | 2000 | lr: 0.03; alpha: 0.1; beta: 0.5; |
| DER++ | 5000 | lr: 0.03; alpha: 0.1; beta: 0.5; |
| ER-ACE | 2000 | lr: 0.03; mom: 0 wd: 0 |
| ER-ACE | 5000 | lr: 0.03; mom: 0 wd: 0 |
| GSS | 2000 | lr: 0.03; |
| GSS | 5000 | lr: 0.03; |
| A-GEM | 2000 | lr: 0.03; |
| A-GEM | 5000 | lr: 0.03; |
| RPC | 2000 | lr: 0.03; |
| RPC | 5000 | lr: 0.03; |
| FDR | 2000 | lr: 0.03; alpha:0.3; |
| FDR | 5000 | lr: 0.03; alpha:0.3; |
| iCaRL | 2000 | lr: 0.03; |
| iCaRL | 5000 | lr: 0.03; |
| BiC | 2000 | lr: 0.03; |
| BiC | 5000 | lr: 0.03; |

## F.3 TASK SEQUENCE DETAILS

In Tables 12, 13 and 14 we report the combination of class order and their division into tasks employed in our experiments during the continual training. Each name corresponds to a different synset of the ImageNet dataset.

Table 12: Mini-ImageNet

| Task | Synsets | | | | |
|------|---------|---------|---------|---------|---------|
| $\tau_1$ | n02091244 | n01770081 | n03207743 | n01749939 | n02110063 |
|  | n02174001 | n02165456 | n02687172 | n09246464 | n02871525 |
|  | n01855672 | n03062245 | n04149813 | n04067472 | n04522168 |
|  | n02138441 | n04509417 | n04275548 | n03888605 | n01981276 |
|  | n02091831 | n03400231 | n02219486 | n02795169 | n03773504 |
|  | n03337140 | n01558993 | n03998194 | n02129165 | n03127925 |
|  | n02457408 | n02108915 | n04389033 | n04604644 | n03908618 |
|  | n02443484 | n02116738 | n03854065 | n03544143 | n09256479 |
|  | n04251144 | n02606052 | n02113712 | n02950826 | n07747607 |
|  | n02108551 | n02108089 | n07613480 | n03527444 | n02823428 |
| $\tau_2$ | n01532829 | n02981792 | n02120079 | n03476684 | n03047690 |
| $\tau_3$ | n02971356 | n02074367 | n06794110 | n04612504 | n03924679 |
| $\tau_4$ | n01910747 | n02105505 | n03584254 | n03770439 | n01930112 |
| $\tau_5$ | n04435653 | n03347037 | n03535780 | n04243546 | n04596742 |
| $\tau_6$ | n02099601 | n04418357 | n02089867 | n03272010 | n03220513 |
| $\tau_7$ | n04146614 | n04443257 | n02111277 | n02747177 | n04515003 |
| $\tau_8$ | n13054560 | n01843383 | n07584110 | n13133613 | n04258138 |
| $\tau_9$ | n03075370 | n02966193 | n03417042 | n03146219 | n03838899 |
| $\tau_{10}$ | n03775546 | n03017168 | n03980874 | n02114548 | n03676483 |
| $\tau_{11}$ | n01704323 | n07697537 | n02101006 | n04296562 | n02110341 |

Table 13: FG-ImageNet

| Task | Synsets | | | | |
|------|---------|---------|---------|---------|---------|
| $\tau_1$ | n01943899 | n01753488 | n01819313 | n01601694 | n01695060 |
|  | n02028035 | n01675722 | n01498041 | n01774750 | n01608432 |
|  | n01685808 | n01978287 | n01537544 | n01742172 | n01924916 |
|  | n01829413 | n01818515 | n01494475 | n01877812 | n02027492 |
|  | n02058221 | n01491361 | n01910747 | n01729977 | n02018207 |
|  | n01824575 | n01986214 | n01860187 | n01773797 | n01630670 |
|  | n01796340 | n01687978 | n01984695 | n01729322 | n01833805 |
|  | n01776313 | n01443537 | n01560419 | n02018795 | n01985128 |
|  | n01677366 | n01755581 | n01739381 | n01770081 | n02013706 |
|  | n01978455 | n02037110 | n01514668 | n01440764 | n01855672 |
| $\tau_2$ | n01756291 | n01770393 | n01775062 | n01632458 | n01820546 |
| $\tau_3$ | n01496331 | n01582220 | n01734418 | n01622779 | n01632777 |
| $\tau_4$ | n01806143 | n01773549 | n01774384 | n02077923 | n01740131 |
| $\tau_5$ | n01484850 | n01914609 | n01665541 | n01667778 | n01847000 |
| $\tau_6$ | n01667114 | n01728572 | n01693334 | n01843383 | n01950731 |
| $\tau_7$ | n01514859 | n02012849 | n01773157 | n01614925 | n01795545 |
| $\tau_8$ | n01944390 | n02011460 | n01883070 | n02002556 | n01798484 |
| $\tau_9$ | n02051845 | n01644900 | n01531178 | n01968897 | n01698640 |
| $\tau_{10}$ | n01592084 | n01955084 | n01930112 | n02007558 | n01735189 |
| $\tau_{11}$ | n01751748 | n01664065 | n01749939 | n02006656 | n01828970 |

Table 14: ImageNet-R

| Task | Synsets | | | | | | | | | |
|------|---------|---|---|---|---|---|---|---|---|---|
| $\tau_1$ | n02165456 | n03594945 | n02325366 | n02814860 | n02966193 | n02480495 | n02106030 | n02088364 | n02066245 | n02843684 |
| | n04591713 | n02110185 | n02092339 | n02980441 | n01833805 | n03947888 | n03602883 | n03649909 | n02841315 | n01855672 |
| | n02007558 | n03424325 | n03710193 | n02992529 | n12267677 | n02233338 | n04254680 | n07714990 | n01644373 | n02077923 |
| | n02138441 | n03498962 | n01484850 | n01847000 | n02113799 | n02129165 | n02119022 | n07697537 | n02480855 | n02009912 |
| | n07693725 | n02445715 | n04487394 | n02802426 | n09835506 | n04133789 | n02113023 | n02091134 | n02110341 | n02317335 |
| | n02607072 | n07768694 | n07880968 | n01843383 | n02769748 | n03494278 | n02106166 | n04086273 | n01944390 | n02098286 |
| | n01806143 | n01514859 | n01498041 | n07614500 | n04465501 | n02398521 | n02117135 | n02808440 | n02112018 | n02906734 |
| | n02486410 | n07720875 | n02110958 | n03124170 | n01632777 | n01986214 | n02437616 | n04192698 | n02134084 | n02655020 |
| | n02086240 | n03345487 | n02395406 | n04147183 | n01748264 | n02113624 | n03272010 | n03495258 | n02128385 | n03467068 |
| | n02096585 | n04310018 | n04146614 | n04536866 | n07745940 | n02088238 | n02363005 | n02364673 | n02226429 | n07753592 |
| $\tau_2$ | n02510455 | n04266014 | n02948072 | n07695742 | n02099712 | n02112137 | n07873807 | n02102318 | n02106662 | n01774750 |
| | n02219486 | n02114367 | n01614925 | n07734744 | n01770393 | n01616318 | n04275548 | n03452741 | n02950826 | n02883205 |
| $\tau_3$ | n02279972 | n03775071 | n01443537 | n02088466 | n04325704 | n02129604 | n02091032 | n07714571 | n02085620 | n04347754 |
| | n01677366 | n04118538 | n01882714 | n07697313 | n01820546 | n02097298 | n02088094 | n03372029 | n02108915 | n02797295 |
| $\tau_4$ | n01531178 | n03930630 | n02268443 | n02823750 | n02106550 | n01494475 | n02190166 | n02346627 | n02130308 | n02481823 |
| | n07718472 | n04522168 | n07753275 | n01910747 | n02447366 | n02109525 | n02099601 | n01784675 | n04141076 | n04389033 |
| $\tau_5$ | n03481172 | n02483362 | n02749479 | n04552348 | n02123045 | n01860187 | n03676483 | n02526121 | n02236044 | n04409515 |
| | n02423022 | n02206856 | n02108089 | n02051845 | n10565667 | n07749582 | n01630670 | n02128757 | n02939185 | n02672831 |
| $\tau_6$ | n02951358 | n07920052 | n01518878 | n02793495 | n03773504 | n01694178 | n09472597 | n02909870 | n02701002 | n03888257 |
| | n01983481 | n02356798 | n02410509 | n07742313 | n02391049 | n03630383 | n02094433 | n02056570 | n02071294 | n01534433 |

