# OpenReview forum: "Dream2Learn: Structured Generative Dreaming for Continual Learning"
_ICLR.cc/2026/Conference — ICLR 2026 Conference Withdrawn Submission_

### Official Review · Reviewer_yfmy · 2025-10-25

**Soundness:** 3
**Presentation:** 4
**Contribution:** 4
**Rating:** 8
**Confidence:** 4

**Summary:**

This paper proposes Dream2Learn (D2L), a generative continual-learning framework inspired by the human dreaming process. Instead of relying on generative-replay past data, D2L uses a latent diffusion model conditioned via soft prompt optimization to synthesize dreamed classes. These dreamed samples form coherent yet distinct new concepts, supporting future task adaptation. Experiments on Mini-ImageNet, FG-ImageNet, and ImageNet-R show consistent gains across rehearsal-based baselines. Ablations validate the role of the oracle and the dynamic dream-class update mechanism.

**Strengths:**

- The paper is well written and easy to follow.
- The proposed method is novel - rather than retrospective replay, D2L introduces a prospective generation mechanism that structures the representation space for future tasks.
- Oracle-guided optimization is an interesting and effective solution to avoid dream collapse.
- Extensive experiments are conducted, including comparisons with SOTA methods and ablation studies. And experimental results demonstrated strong imporvements.

**Weaknesses:**

- The “dreaming vs. replay” distinction is interesting but not sharply formalized, and the boundary seems vague. There're OOD test, but not sufficient, as being “not old classes” doesn’t prove they’re future-oriented or structurally bridging.
- D2L adopt a pretrained diffusion backbone for generating “dreamed classes.” However, how the choice of this generator impacts results is not analyzed.

**Questions:**

- Can you provide quantitative or visual evidence that the dreamed samples truly occupy intermediate latent regions between past and future classes? And the generated samples indeed anticipating future?
- Would wrong anticipation harm the performance in some cases?

---

> ### Author Response · Authors · 2025-11-22
> **Response to Reviewer yfmy (1/2)**
>
> We thank the reviewer for the review and the detailed feedback.
>
> >**[W1] The “dreaming vs. replay” distinction is interesting but not sharply formalized, and the boundary seems vague. There're OOD test, but not sufficient, as being “not old classes” doesn’t prove they’re future-oriented or structurally bridging.**
>
> Thank you for the comment. The distinction between “dreaming” and “replay” in continual is nuanced and not yet sharply formalized in literature. While we attempt to conceptually frame dreaming as the generation of novel, intermediate classes to aid transfer rather than mere replay of old-class samples, we acknowledge that the boundary remains, to some extent, heuristic.
>
> Our use of out-of-distribution (OOD) test benchmarks aims to provide some evidence that dreamed classes facilitate generalization beyond replaying old classes. However, we concur that demonstrating future-oriented or structurally bridging properties of dreamed classes comprehensively is a significant open challenge.
>
> Currently, our strongest evidence rests on the improved forward transfer (FWT) metrics and ablation studies that show dreaming-induced benefits beyond traditional replay. The OOD tests complement this by illustrating performance in more challenging, shifted domains.
>
> However, structural coherence and semantic distinguishability are central to our framework and have emerged empirically through our experiments. Semantic distinguishability is evident via clear inter-class differences, and structural coherence through recurring intra-class visual patterns (e.g., yellow-red spikes for buses, green-black-red motifs for ladybugs).
>
> To quantitatively assess structural coherence, we extracted deep features from dream samples using AlexNet, clustered them using K-Means with the number of clusters equal to the dream classes, and computed normalized mutual information (NMI) and adjusted Rand index (ARI) with respect to ground-truth labels. We obtained an average NMI of 0.6501 and ARI of 0.4919, providing quantitative evidence supporting meaningful class-aligned structure in the dreamed samples in addition to the visual patterns observed. While these scores are not perfect, as expected in generative settings, they reinforce the notion that dreaming yields structurally coherent and semantically interpretable representations.
>
> >**[W2] D2L adopt a pretrained diffusion backbone for generating “dreamed classes.” However, how the choice of this generator impacts results is not analyzed.**
>
> D2L, is in principle, compatible with any image generator capable of class-conditional generation via prompts, with the only requirement being the ability to backpropagate the classifier’s loss through the entire generation pipeline, from the classifier output to the prompt being optimized.
>
>
> We have observed that quality of generation has an impact on D2L’s performance. For instance, with fixed prompt optimization (Table 4), prompt collapse leads to a sharp drop in accuracy (38.94%). Using real images as dreams (as in WSCL) yields 42.38 ± 1.16%, while our full D2L approach with LDM and oracle-guided adaptive stopping achieves 40.90 ± 0.95%. This demonstrates how D2L strikes a balance between sample quality and practical considerations.
> Exploring lighter or alternative generators is certainly a promising direction, but in our view, preserving sample diversity and maintaining proper class alignment, thereby avoiding prompt collapse, is the most critical requirement.
>
> >**[Q1]Can you provide quantitative or visual evidence that the dreamed samples truly occupy intermediate latent regions between past and future classes? And the generated samples indeed anticipating future?**
>
>
> Thank you for the question. To provide visual evidence, we computed t-SNE projections involving the new real class, the corresponding dreamed class selected by D2L, and the most similar real class from the previous task. The t-SNE image is available **[here](https://ibb.co/0j8yJBkL)**.
>
> In this visualization:
>
> - *real_4* is the real class from the new incoming task,
> - *dream_4* is the dreamed class chosen by D2L to initialize that class,
> - *task0_dream4* is the most similar real class from the previous task.
>
>
> As it can be noticed, the dreamed samples form a separate, coherent cluster that is:
> distinct from both the previous and upcoming real classes,
> yet closer to the new real class than the closest class from the previous task,
> thus anticipating the future distribution without leaking it.
>
>
> This pattern holds consistently in our multi-class t-SNE as well (**[multi-class t-SNE image](https://ibb.co/pBthH7KM)**): dreamed samples cluster separately, and dream cluster generally lies nearer to its corresponding future real class than to any past class, even though the visualization becomes more crowded.
>
> We will include these analysis in the supplementary material and comment on them in the revised version.

---

> ### Author Response · Authors · 2025-11-22
> **Response to Reviewer yfmy (2/2)**
>
> > **[Q2] Would wrong anticipation harm the performance in some cases?**
>
> We did not observe cases where dreamed samples misled the classifier in a way that caused performance degradation. Even in scenarios where trajectory collapse occurred (as illustrated in Figure 4 of the paper), this resulted in no improvement, but not in harmful effects on classifier performance.
>
> However, we did observe misleading behavior in specific settings where the classifier backbone overfits and fails to learn robust representations. In such cases, the generator tends to collapse into trivial or spurious representations that closely overlap with those of previously seen classes. This leads to ambiguous or shortcut-driven generations, which in turn degrade performance.

---

> ### Comment · Reviewer_yfmy · 2025-11-25
>
> Thank the authors for the response. I have reviewed the author's response and other reviewers' comments. First, the proposed method is novel - rather than retrospective replay, D2L introduces a prospective generation mechanism that structures the representation space for future tasks. However, I agree with reviewer HqT1 that comparison with a full replay [a] method on the amount of compute (e.g. total time spent in the training phase, FLOP etc.) and memory are necessary.

---

### Official Review · Reviewer_m5Na · 2025-10-30

**Soundness:** 3
**Presentation:** 3
**Contribution:** 1
**Rating:** 4
**Confidence:** 3

**Summary:**

The paper develops D2L based on generative replay to address catastrophic forgetting in CL. The core idea is to enable a base model to generate structured data synthetically that are semantically distinct yet structurally coherent with previously learned knowledge to enable experience replay. Unlike past work that mainly reconstruct input data, D2L leverages a latent diffusion model. This model is conditioned on soft prompt optimization to create future-adaptive representations for experience replay. This data expands the classifier’s latent space and enable forward transfer to improve generalization on new tasks. As a result, the process does not need external supervision or additional real data. During training, D2L optimizes prompts for each learned class to synthesize new classes, guiding the generator toward distinct but consistent outputs. An oracle network monitors the optimization process to prevent collapse and overfitting. Unlike memory buffer-based methods, D2L does not store generated data but maintains an inventory of optimized prompts which are easier to store. Empirical evaluation on three benchmarks demonstrates that D2L outperforms existing baselines. The paper also provides ablative experiment to demonstrate the role of the oracle in maintaining sample diversity, and the superiority of D2L’s structured data generation over interpolation-based alternatives.

**Strengths:**

1. The paper is well organized and can be read straightforwardly.

2. The method and the experiments consider forward transfer which is overlooked in many CL works, yet is very important in CL.

3. D2L introduces an oracle network that learns when to stop the soft prompt optimization. This idea is novel and to my knowledge unexplored in previous works.

4. Experimental setup is sound and demonstrate that D2L is effective.

**Weaknesses:**

1. Addressing catastrophic forgetting based on generative replay is a relatively old idea in CL, including several works not referenced in the paper, and hence the novelty of this work is limited. It is true that implementation of this idea is new but the core idea is not mew.

2. D2L relies on a diffusion model which is a large model augmented to the base ResNet-like classifier. This addition makes the model far more complex and given the scope of experiments, one can argue just to use several ResNets, one per task, to get even better performance results.

3. Evaluations include only 10–11 tasks, each with a small number of classes. It’s unclear how the method performs when more tasks are used.

4. The baselines that were used for comparison are mostly old baselines. It is OK to include them but comparisons need to expand to include all methods of the past three years to demonstrate competitive performance.

5. The benchmarks that are used are on the simpler side of CL benchmarks at the moment and a relatively old model is used in experiments. Experiments should include more recent benchmarks, e..g, CLEAR or CLAD.

6. The code is not available which makes judgement about reproducibility of the results challenging.

**Questions:**

1. Why backward transfer is not reported in the experiments? In CL, it is as important as forward transfer. I understand accuracy reflects that to some extend but still it is an independent yet crucial in CL.

---

> ### Author Response · Authors · 2025-11-21
> **Response to Reviewer m5Na (1/3)**
>
> We thank the reviewer for the review and the detailed feedback.
>
> > **[W1] Addressing catastrophic forgetting based on generative replay is a relatively old idea in CL, including several works not referenced in the paper, and hence the novelty of this work is limited. It is true that implementation of this idea is new but the core idea is not new.**
>
> We clarify that D2L is not a generative replay (GR) in any sense. GR methods train a generator to reconstruct past data and use those reconstructions to replace or reduce the replay buffer. D2L does none of this:
> 1) the diffusion model is frozen, never trained on the stream;
> 2) it never attempts to regenerate past samples;
> 3) it does not replace or shrink the replay buffer;
> 4) and the generated samples are new, intermediate classes, not reconstructions.
>
> The goal of D2L is to reshape the representation space to improve future-task learning, not to replay previous tasks. To the best of our knowledge, no prior CL work has proposed generating synthetic intermediate classes, guided by the model’s own evolving knowledge, to enhance forward transfer. This mechanism is new and distinct from generative replay.
>
> > **[W2] D2L relies on a diffusion model which is a large model augmented to the base ResNet-like classifier. This addition makes the model far more complex and given the scope of experiments, one can argue just to use several ResNets, one per task, to get even better performance results.**
>
> To clarify a central misunderstanding: the diffusion model is never part of the continual learner itself.
>
> During training, it is used only as a frozen external generator to produce intermediate synthetic samples. It is never fine-tuned, never updated with stream data, and never integrated into the classifier, only a small soft prompt is learned in order to to produce new class-specific synthetic images that help stabilize loss dynamics and regularize the transition when a new task arrives
> During inference, the diffusion model is not used at all. The deployed model is exactly a single ResNet-18, identical in size and computational cost to all baselines.
> To avoid confusion, we clearly distinguish the two phases of D2L, as common for any CL method:
>
> 1) During training, the diffusion model is used only as a frozen external generator to create synthetic samples. It is never fine-tuned, never updated with stream data, and not integrated into the classifier. The only learned component on the generative side is a very small soft prompt, whose role is to produce new class-specific synthetic images that help stabilize loss dynamics and regularize the transition when a new task arrives.
>
> 2) During inference, the diffusion model is not used at all. At test time, D2L is strictly a standard ResNet-18 model, identical to the baselines in size, parameter count, and computational cost. No generative component remains in the deployed model.
>
> Regarding the suggestion to “use several ResNets, one per task”: while this is technically possible, it is not a meaningful comparison for our setting. Such an approach would require keeping multiple full models at inference time, multiplying the parameter count and, crucially, relying on knowledge of the task identity (oracle condition). A multi-model oracle setup sidesteps the central challenge of the class-incremental learning setting we tackle, namely, how a single evolving model adapts to a non-stationary stream while retaining past knowledge.
>
> > **[W3] Evaluations include only 10–11 tasks, each with a small number of classes. It’s unclear how the method performs when more tasks are used.**
>
> Our current experiments focus consistently with the established protocols and popular dataset in the CL literature, such as Mini-ImageNet, FG-ImageNet, and ImageNet-R. This choice allows us to evaluate the key properties of D2L in well-understood scenarios and ensures comparability with prior works.
>
> It is important to note that our D2L is designed to be equipped on top of an existing CL method, with the dreaming process guided by the continual classifier. As is common in class-incremental learning, increasing the number of tasks typically leads to a gradual decrease in final average accuracy, reflecting the growing challenge of discriminating among a larger number of tasks and classes.

---

> ### Author Response · Authors · 2025-11-21
> **Response to Reviewer m5Na (2/3)**
>
> > **[W4] The baselines that were used for comparison are mostly old baselines. It is OK to include them but comparisons need to expand to include all methods of the past three years to demonstrate competitive performance.**
>
> We understand the suggestion to broaden the comparisons. We note, however, that our experimental protocol follows the standard training-from-scratch CNN regime adopted in rehearsal-based class-incremental learning. Many continual-learning methods from the past three years rely on large ViT backbones pre-trained on ImageNet-(1K/21K) or on prompt-tuning frameworks, which operate in a different evaluation regime and are therefore not directly comparable: their performance is strongly influenced by pretraining and architectural inductive biases rather than by the CL strategy itself (beside requiring necessarily a vision transformer).
>
> For fairness and methodological consistency, we compare only with methods that (i) use the same backbone, (ii) are trained from scratch, (iii) operate in single-head class-incremental settings, and (iv) employ replay, since D2L interacts directly with the buffer. Within this controlled and widely adopted regime, we include strong and representative baselines (ER, ER-ACE, DER++, GSS, RPC, BiC), which remain the standard benchmarks for rehearsal-based CL.
>
> > **[W5] The benchmarks that are used are on the simpler side of CL benchmarks at the moment and a relatively old model is used in experiments. Experiments should include more recent benchmarks, e..g, CLEAR or CLAD.**
>
> We would appreciate clarification regarding which specific configurations of CLEAR or CLAD the reviewer is referring to, as both frameworks include multiple variants and evaluation protocols with assumptions that **differ substantially from our setting**. In particular, CLAD (Poggi et al., 2022) is designed for autonomous-driving scenarios and includes two tracks: CLAD-C, a streaming classification benchmark with temporal and domain drift, and CLAD-D, an object-detection benchmark. Neither corresponds to the class-incremental classification setup we study, where tasks introduce disjoint new classes, evaluation is single-head, and models are trained from scratch in a rehearsal-based setting. Similarly, CLEAR focuses on real-world streaming with chronological drift rather than class-incremental task boundaries.
>
> Our paper evaluates D2L on three established and diverse Class Incremental Learning benchmarks, Mini-ImageNet, FG-ImageNet, and ImageNet-R, which represent the standard evaluation protocol used in recent rehearsal-based CIL work.
>
> > **[W6] The code is not available which makes judgement about reproducibility of the results challenging.**
>
> Thanks for the question. As described in the paper, the continual learning components of our work are implemented using the Mammoth framework, which is publicly available and widely adopted by the community. This ensures that the core continual learning evaluation is grounded in a reliable codebase. The novel parts of our contributions—namely, the prompt optimization, image generation and class assignment modules- are developed specifically for this work.
>
> We plan to release the full implementation, including all new modules and training pipelines, upon acceptance, allowing us to incorporate feedback from the review process and ensure a clean, well-documented release.

---

> ### Author Response · Authors · 2025-11-21
> **Response to Reviewer m5Na (3/3)**
>
> > **[Q1] Why backward transfer is not reported in the experiments? In CL, it is as important as forward transfer. I understand accuracy reflects that to some extend but still it is an independent yet crucial in CL.**
>
> Thank you for the question. We computed the BWT for all methods and datasets, with results as reported in the table below:
>
> | BWT           | Mini-ImageNet        |          | FG-ImageNet          |          | ImageNet-R           |          |
> |---------------|---------------------|----------|----------------------|----------|----------------------|----------|
> |               | 2000                | 5000     | 2000                 | 5000     | 2000                 | 5000     |
> | ER            | -44.01 ± 3.71       | -33.04 ± 5.45 | -48.45 ± 2.58       | -42.72 ± 1.99 | -22.97 ± 1.82       | -21.15 ± 1.33 |
> | ER + D2L      | -53.38 ± 3.15       | -41.18 ± 3.11 | -57.81 ± 3.52       | -43.41 ± 4.44 | -21.28 ± 1.05       | -17.53 ± 2.26 |
> | DER++         | -41.69 ± 7.18       | -36.99 ± 5.87 | -41.12 ± 8.01       | -32.49 ± 5.66 | -25.74 ± 1.00       | -23.69 ± 2.69 |
> | DER++ + D2L   | -62.72 ± 5.47       | -53.58 ± 5.27 | -58.11 ± 3.31       | -55.88 ± 5.67 | -26.41 ± 1.61       | -25.68 ± 2.81 |
> | ER-ACE        | -4.91 ± 4.38        | -1.98 ± 4.78  | -3.24 ± 4.36        | -0.64 ± 11.09 | -2.27 ± 0.45        | -1.01 ± 0.58  |
> | ER-ACE + D2L  | -23.02 ± 3.63       | -11.93 ± 4.07 | -17.19 ± 1.76       | -11.84 ± 2.09 | -4.25 ± 1.99        | -0.47 ± 1.53  |
>
>
> These results reinforce the narrative we discussed earlier about the stability-plasticity trade-off inherent in continual learning. Specifically, adding D2L tends to increase forgetting - as measured by more negative BWT, particularly in the replay-centric baselines ER and DER++. This reflects increased plasticity, allowing the model to adapt and incorporate new knowledge more readily.
>
> However, for ER-ACE, which already maintains a relatively low forgetting baseline, D2L also increases forgetting moderately, but the absolute magnitude remains small. Importantly, as we emphasized before, this increased forgetting with D2L is accompanied by substantial gains in forward transfer and overall final accuracy, indicating a beneficial shift in the stability-plasticity balance.
>
> Therefore, BWT should be interpreted in the broader context of continual learning performance metrics: a moderate increase/decrease in forgetting/BWT may be acceptable, or even desirable, if it facilitates improved learning of new tasks and knowledge transfer.
>
> We will include these metrics in the supplementary material and comment on them in the revised version.

---

> ### Author Response · Authors · 2025-11-21
> **Request for Clarification on Score Change**
>
> We would like to ask the reviewer for clarification regarding the unexplained lowering of the score for our paper, which occurred approximately 12 hours after the reviews were released and without any accompanying comment or justification. According to the ICLR guidelines:
>
> > “Reviewers can change their reviews after the author response period. They should update their review and final recommendation taking into account the discussion phase, and clearly state what did/didn’t change their recommendation.”

---

> > ### Comment · Reviewer_m5Na · 2025-11-24
> > **Post-rebuttal update**
> >
> > Thank you for answering the questions. After reading the responses and the rest of reviews, I decided to maintain my initial rating. I share the primary concerns of Reviewer HqT1, as noted in my initial review. I don't find the response convincing and hence don't think this paper is above the acceptance bar.

---

> > > ### Comment · Area_Chair_vyUM · 2025-11-28
> > > **Rationale for change in rating**
> > >
> > > Dear Reviewer m5Na,
> > > Many thanks for your time and effort reviewing this paper, and for taking the time to respond to the author rebuttal.
> > > In your response to the author rebuttal, you indicate that you "decided to maintain [your] initial rating". However, as the authors point out in their response, your initial rating when submitting your review was 4, while your current rating is 2.
> > > Would it be possible to provide a rationale for this change in your recommendation? Given that this paper currently has quite divergent recommendations (two reviewers recommending acceptance, two reviewers recommending rejection), such a rationale would also be very helpful for arriving at a decision for this paper.
> > > Many thanks.
> > > AC

---

### Official Review · Reviewer_wzqW · 2025-10-31

**Soundness:** 3
**Presentation:** 2
**Contribution:** 3
**Rating:** 6
**Confidence:** 3

**Summary:**

This paper proposes a conditional generative replay framework to improve forward transfer in class-incremental learning (CIL). Instead of using generative replay to combat forgetting, the method aims to improve the learning of new classes. Specifically, the authors condition a latent diffusion model on a random image in the replay buffer and one of the previously seen classes to generate novel "dreamed" images. These generated samples are assigned to new pseudo-classes and are used, together with incoming and buffered images, to train the encoder/classifier. The resulting class embeddings are then used to initialize the embeddings of new real classes encountered later, which is argued to promote forward transfer. Experiments on ImageNet-based CIL benchmarks show consistent improvements over buffer replay baselines. Ablation studies show the contribution of each proposed component.

Overall, I think this is an interesting paper that rethinks the role of generative replay in continual learning. The empirical improvements are convincing, and the idea of leveraging dreamed classes for forward transfer is novel. My major concerns are related to the clarity of the presentation and additional analysis on the scaling properties, forgetting behavior, and sources of improvement.

**Strengths:**

- To my knowledge, the central idea of using generative replay for forward transfer rather than mitigating forgetting is novel and interesting.
- Experiments and ablations show the effectiveness of the proposed method.

**Weaknesses:**

1. The method involves multiple class mappings and set operations, and their description lacks clarity (see questions below). It might be helpful to point each operation in Algorithm 1 to the sections or equations where that operation is described.
2. The method seems to rely on several class embedding manipulations, and the experiments are performed using ResNet-18. It is not clear what challenges might arise when scaling up this approach to larger data or to settings where classes are not disjoint between tasks. Do the authors have any thoughts on this?
3. The method introduces several new modules, but only the buffer size is controlled when comparing different methods. A time-efficiency analysis might be helpful.
4. Forgetting is not reported. This could be informative---e.g., does the model trade off some forgetting for forward transfer?
5. It is unclear to me whether FWT comes from augmenting past classes with generated images (leading to more robust representations) or from reusing class embeddings for new classes. A helpful baseline would assign the dreamed samples to their conditioning classes without reusing class embeddings for initialization.

**Questions:**

1. I didn’t understand why you had a mechanism to map dreamed classes to classes in the new task (L175) to improve FWT, but then state that the dreamed classes do not reflect unseen classes (Sec. 4.3). Aren’t these statements conflicting? Why do we not want them to reflect unseen classes?
2. L285: Does "feature embeddings extracted by the classifier" refer to the features before the last layer? Is $f$ the encoder and $F$ encoder+classifier head?
3. The Oracle uses $\mathbf{Z}_t$ as input features, but I can’t find how the stopping decisions, which are used as targets, are labeled.

### Questions/comments that did not impact the score
4. Please fix the spacing between paragraphs.
5. How did you visualize the latent space in 2D in Figure 3?
6. The paper emphasizes that prior work is not bio-plausible (e.g., L59, L107, L442), but dreams in the brain occur late in the visual cortex [Hor+13], suggesting that the proposed method, which generates pixels directly, is also not bio-plausible. I would adjust the narrative accordingly.
7. L247: Should it be "$c^{\text{out}}$ replaces the dream classes" instead?
8. The end of Sec. 4.2 could be moved to the related work section.
9. L437: This sentence should be a hypothesis rather than being directly supported by Table 5.
10. L442: Why is WSCL not listed in the table, and what is a “true dreaming process”?


[Hor+13] Neural decoding of visual imagery during sleep. Horikawa et al., 2013. Science.

---

> ### Author Response · Authors · 2025-11-21
> **Response to Reviewer wzqW (1/4)**
>
> We thank the reviewer for the review and the detailed feedback.
> > **[W1] The method involves multiple class mappings and set operations, and their description lacks clarity (see questions below). It might be helpful to point each operation in Algorithm 1 to the sections or equations where that operation is described.**
>
> We agree that Algorithm 1 could be clarified further. We will soon update the submission file where we will explicitly link each operation in Algorithm 1 to the corresponding section or equation where it is defined or discussed. This should make the flow of the method easier to follow and reduce ambiguity in the notation.
>
> > **[W2] The method seems to rely on several class embedding manipulations, and the experiments are performed using ResNet-18. It is not clear what challenges might arise when scaling up this approach to larger data or to settings where classes are not disjoint between tasks. Do the authors have any thoughts on this?**
>
> In this work we adopt ResNet-18 and standard class-incremental benchmarks to align with the predominant experimental protocol in the CL literature and to isolate the effect of the proposed dreaming mechanism from that of large backbones. D2L is backbone-agnostic, with its core operations taking place at the classifier head and within the latent space of the frozen diffusion model, making minimal assumptions about the backbone design.
>
> With respect to scaling to larger data and more classes, the computational cost of our approach grows linearly with the number of classes, as each new real class requires optimizing a corresponding dream prompt. This procedure is conceptually simple and scales naturally as the number of classes increases.
>
> Regarding settings with non-disjoint classes between tasks, D2L does not fundamentally require strict class separation. Each semantic class is linked to (i) a fixed output neuron for real data and (ii) a set of dream neurons and learnable prompts. These associations remain valid even if the same class reoccurs in multiple tasks or under different domains, as long as its semantic identity (label) is consistent. In such blurred-boundary scenarios, incoming samples of an already seen class refine both the real-class representation and dream prompts, with the oracle mechanism continuing to regulate the optimization process for each dream class. We thus anticipate no conceptual obstacles to applying D2L in settings with non-disjoint classes; rather we see this as a natural extension of our framework.
>
> > **[W3] The method introduces several new modules, but only the buffer size is controlled when comparing different methods. A time-efficiency analysis might be helpful.**
>
> D2L adds overhead only during training due to the optimization of dream prompts and the evaluation of the oracle-based stopping criterion. Importantly, there is no additional cost at inference time: all methods, including D2L variants, deploy exactly the same ResNet-18 classifier without any dependence on additional modules. Therefore, test-time latency and memory footprint remain identical across baselines.
>
> To provide a more concrete insight into the training overhead, we measured the average wall-clock time per optimization step on an Nvidia A6000 GPU by repeating a single step 100 times:
> Prompt optimization: 0.72 ± 0.04 seconds
> Computing Oracle metrics: 1.83 ± 0.05 seconds
> Generating a batch of images: 1.89 ± 0.02 seconds
> In total, a full optimization step takes approximately 4.5 seconds. With an average oracle stopping time around 100 steps, optimizing a single dream class requires roughly 7.5 minutes. Again, this overhead incurs at training time only: all baselines are equally efficient at inference time.

---

> ### Author Response · Authors · 2025-11-21
> **Response to Reviewer wzqW (2/4)**
>
> > **[W4] Forgetting is not reported. This could be informative---e.g., does the model trade off some forgetting for forward transfer?**
>
> The reviewer’s intuition is right: D2L is explicitly designed to increase plasticity, and therefore an increase in forgetting is an expected effect of the method. Indeed, the goal of D2L is to enhance forward transfer by enabling the model to reorganize its representations for future tasks, and this necessarily shifts the stability-plasticity balance. To make this clear, we report average forgetting for ER, DER++, and ER-ACE and their D2L variants across all benchmarks and buffer sizes:
>
> | Forgetting      | Mini-ImageNet 2000 | Mini-ImageNet 5000 | FG-ImageNet 2000 | FG-ImageNet 5000 | ImageNet-R 2000 | ImageNet-R 5000 |
> |-----------------|--------------------|--------------------|------------------|------------------|-----------------|-----------------|
> | ER              | 44.01±3.71         | 33.04±5.45         | 48.45±2.58       | 42.72±1.99       | 22.97±1.82      | 21.15±1.33      |
> | ER + D2L        | 53.38±3.15         | 41.18±3.11         | 57.81±3.52       | 43.41±4.44       | 21.28±1.05      | 17.53±2.26      |
> | DER++           | 41.84±7.18         | 37.24±5.59         | 41.25±8.15       | 33.00±5.20       | 25.74±1.00      | 23.69±2.69      |
> | DER++ + D2L     | 62.72±5.47         | 53.58±5.27         | 58.11±3.31       | 55.88±5.67       | 26.41±1.61      | 25.68±2.81      |
> | ER-ACE          | 10.12±4.04         | 5.99±3.42          | 8.60±4.95        | 8.44±9.09        | 2.92±0.45       | 2.44±0.57       |
> | ER-ACE + D2L    | 23.43±3.44         | 12.40±4.13         | 17.36±1.73       | 13.25±1.58       | 4.73±1.45       | 2.91±1.05       |
>
>
> The results confirm the reviewer’s point: for replay-only baselines such as ER and DER++, D2L increases forgetting while simultaneously producing substantial gains in FAA and consistently positive forward transfer (Table 2). Overall, the global FAA metric shows that the benefits introduced by D2L overcome its effects on forgetting.   For instance, for ER-ACE, which is engineered to keep forgetting extremely low, D2L raises forgetting only moderately while still improving FAA and turning FWT from negative to positive.
>
> We will include forgetting metrics in the supplementary material and explicitly discuss this expected trade-off in the revised version.
>
> > **[W5]  It is unclear to me whether FWT comes from augmenting past classes with generated images (leading to more robust representations) or from reusing class embeddings for new classes. A helpful baseline would assign the dreamed samples to their conditioning classes without reusing class embeddings for initialization.**
>
> We clarify that positive forward transfer in D2L does not arise from augmenting past classes with synthetic images, but from creating semantically shifted intermediate classes that reorganize the representation space for future tasks.
> As shown in our experiments (see responses to Rev. HqT1), using the diffusion model as a generic data augmentor does not improve continual learning:
>
> a) When real buffer samples are removed and rehearsal relies entirely on generated images, performance collapses (e.g., Mini-ImageNet FAA drops to 6-9%).
>
> b) When BiC is trained with the same number of diffusion-generated samples used by D2L, accuracy decreases on Mini-ImageNet and FG-ImageNet relative to BiC without synthetic augmentation.
>
> These results show that augmenting past classes with diffusion samples is insufficient to produce positive forward transfer and can even be detrimental due to distribution shift. Thus, FWT cannot simply come from “adding more images” to past classes.
>
> As for the reviewer's proposed baseline, i.e., assigning dreamed samples to their conditioning classes without reusing class embeddings for initialization, this approach fundamentally contradicts our dreaming mechanism.  During prompt optimization, we explicitly halt the refinement of a dream as soon as the generated image is classified with a label different from the conditioning label, ensuring that it represents a semantically shifted, intermediate concept rather than the original class. Assigning such a dream back to its conditioning class would effectively introduce deliberate label noise, affecting model’s performance on real data.

---

> ### Author Response · Authors · 2025-11-21
> **Response to Reviewer wzqW (3/4)**
>
> > **[Q1] I didn’t understand why you had a mechanism to map dreamed classes to classes in the new task (L175) to improve FWT, but then state that the dreamed classes do not reflect unseen classes (Sec. 4.3). Aren’t these statements conflicting? Why do we not want them to reflect unseen classes?**
>
> We appreciate the reviewer’s question and agree that the two statements may appear contradictory at first sight. In reality, they refer to two different aspects of the dreaming mechanism:
>
> (1) Why we map dreamed classes to new real classes.
>
> Dreamed classes are designed to provide intermediate feature prototypes that can be reused to initialize new real classes. This improves forward transfer without requiring the dream to resemble the future class itself. Their role is to offer useful structure, not to predict the semantic identity of upcoming tasks.
>
> (2) Why we state that dreamed classes do not reflect unseen classes.
>
> This clarification is meant to rule out a different concern: because we use a pretrained diffusion model, one might suspect that it could inadvertently generate samples that resemble future classes, leading to leakage, which we explicitly quantify and discuss in Section 5 (Limitations). Our qualitative inspection and quantitative leakage test both confirm that the dreamed samples do not semantically match future classes.
>
> Thus, there is no conflict:
>
> – Dreamed classes should be structurally helpful for adapting to new tasks.
>
> – They should not reproduce or anticipate the semantics of future classes (to avoid leakage).
>
> > **[Q2] L285: Does "feature embeddings extracted by the classifier" refer to the features before the last layer? Is $f$  the encoder and $ F$ encoder+classifier head?**
>
> In this context, “feature embeddings extracted by the classifier” refers specifically to the representations output by the encoder, i.e. the features before the final classification head. We will clarify this point explicitly in the revised version to avoid any ambiguity.
>
> > **[Q3] The Oracle uses $Z_t$ as input features, but I can’t find how the stopping decisions, which are used as targets, are labeled.**
>
> Thank you for the question. The stopping decisions used to train the oracle are obtained through manual annotation of dream trajectories. For each dreamed class, we generate the full sequence of intermediate representations across optimization steps and then manually mark the iteration at which further refinement no longer improves the semantic shift or usefulness of the dream. This yields a binary target indicating whether refinement should continue or stop at each step of the trajectory.
>
> The oracle network takes as input the aggregated feature vectors from successive optimization steps (as described in Section 3.2 and Appendix B) and learns to predict the stopping decision from this labeled data.
> We observe that the oracle generalizes reliably: its predicted stopping points differ from the manually annotated ones by only 9.28 steps on average, and these differences produce no measurable impact on continual learning performance. Across all benchmarks, the oracle produces stable stopping behavior and we did not observe trajectory collapse.
>
> > **[Q5] How did you visualize the latent space in 2D in Figure 3?**
>
> The 2D visualization in Figure 3 is intended as an illustrative conceptual support rather than an empirical projection of the actual latent space.
>
> > **[Q6] The paper emphasizes that prior work is not bio-plausible (e.g., L59, L107, L442), but dreams in the brain occur late in the visual cortex [Hor+13], suggesting that the proposed method, which generates pixels directly, is also not bio-plausible. I would adjust the narrative accordingly.**
>
> Thank you for the observation.  Our intention was not to claim biological plausibility at the level of the generative mechanism, but rather to draw on the functional inspiration that internally generated activity can support learning and adaptation. As noted by Horikawa et al. (2013), biological dreams emerge from high-level cortical representations rather than pixel-level imagery; in that sense, D2L is not meant to mirror the biological process. We will clarify this distinction in the revised manuscript to better align the narrative with current neuroscience understanding.

---

> ### Author Response · Authors · 2025-11-21
> **Response to Reviewer wzqW (4/4)**
>
> > **[Q7] L247: Should it be " replaces the dream classes" instead?**
>
> Thank you for pointing this out. We understand that the original sentence may be unclear. In the revised version of the paper, we will rephrase it for improved clarity as follow:
>
> “In practice, each new dream class $c^d$ replaces a previous dream class with which it is most closely aligned in terms of classification likelihood. “
>
> > **[Q8]The end of Sec. 4.2 could be moved to the related work section.**
>
> Thank you for the suggestion. We will move the end of Section 4.2 accordingly in the revised version (which will be updated soon) of the paper.
>
> > **[Q9]L437: This sentence should be a hypothesis rather than being directly supported by Table 5.**
>
> We agree that the sentence can be better framed as a hypothesis rather than a definitive conclusion.
>
> To clarify, in the “incremental dreaming” setting, where new dream classes are introduced at every task, the total number of classes (real and dreams) that the model is trained on effectively increases over time. This expansion naturally complicates the classification task, as the model must distinguish among a larger set of classes during training, yet the evaluation is always performed exclusively on the real classes. It is reasonable to hypothesize that this increasing label space could impact performance, making the task more challenging, and this perspective will be reflected more explicitly in the revised version of the paper.
>
> > **[Q10]L442: Why is WSCL not listed in the table, and what is a “true dreaming process”?**
>
> Thank you for the question. Although WSCL does not appear in the main results table, we do compare D2L against WSCL In Section 4.3 (Model Analysis). We chose not to include WSCL in the main table because it relies on an auxiliary real dataset to pre-activate future task heads, effectively providing additional information that is not available to the continual learner in the standard class-incremental setup. Including it in the main results table would therefore conflate differences in data access rather than differences in method quality.
>
> Regarding the phrasing “true dreaming process,” our intention was simply to highlight this distinction:
>
> – WSCL relies on external real images from an auxiliary dataset to guide its sleep phase.
>
> – D2L generates internal synthetic classes directly from the model’s own knowledge and a frozen diffusion prior.
>
> To avoid confusion, we will revise the wording. Our meaning was not to devalue WSCL, but to clarify that D2L uses internally generated, classifier-guided samples without any auxiliary dataset

---

### Official Review · Reviewer_HqT1 · 2025-11-01

**Soundness:** 2
**Presentation:** 2
**Contribution:** 1
**Rating:** 0
**Confidence:** 5

**Summary:**

The paper proposes Dream2Learn (D2L), a generative-replay approach for continual learning (CL) that conditions a diffusion model via soft-prompt optimization to generate semantically distinct and structurally coherent dreamed classes. The dreamed samples are interleaved with training to regularize representations to reduce forgetting. The method results in positive forward transfer on Mini-ImageNet, FG-ImageNet, and ImageNet-R.

**Strengths:**

1) The paper provides an interesting motivation based on human sleep based replay for CL.
2) Results on standard benchmarks demonstrate superior performance of D2L compared to other replay based methods.

**Weaknesses:**

1) While catastrophic forgetting is an important problem to consider in CL, the main reason to not do full replay of prior data to avoid forgetting is to reduce the amount of prior data storage, and additional training on prior data to save compute. The proposed method uses significantly larger models to generate more classes to train a small CNN on a relatively small dataset (ImageNet-100). Not only does the model need to train on a large number of generated class images, the method still relies on partial replay of prior data, leading to a much higher compute usage. Additionally, the generative models seemed to have already been trained on huge amount of data, which already covers simple datasets used for CL (e.g. ImageNet-100). In this case, why can't we just use large foundation models as zero shot classifiers on these datasets? Why can't we use full replay of prior data rather than spending compute on generating data with large diffusion models? How do you ensure no leakage from the diffusion model on the ImageNet datasets.
2) There have been many feature replay based techniques that reduce forgetting without spending a signifincant amount of compute. Have the authors considered comparing to those methods?
3) There is only a small gain in accuracy compared to partial replay methods, such as BiC. Note that BiC does not use any extra generative or pre-trained models, and only relies on a small buffer of prior data. For a fair comparison, it is essential that the other models are provided with similar amount of compute/data.

Given these concerns, the contribution of this paper to continual learning is minimal.

**Questions:**

Please see the weaknesses section for my questions.

---

> ### Author Response · Authors · 2025-11-21
> **Response to Reviewer HqT1 (1/3)**
>
> We appreciate the reviewer’s comments and address below a few key misunderstandings regarding the performance gains, the role of synthetic data, and the fairness of our comparison.
>
> > **[W1a] While catastrophic forgetting is an important problem to consider in CL, the main reason to not do full replay of prior data to avoid forgetting is to reduce the amount of prior data. [...] Why can't we use full replay of prior data rather than spending compute on generating data with large diffusion models?**
>
> We disagree with the assumption that continual learning is primarily motivated by memory constraints. This is not supported by the CL literature. **The foundational motivation of continual learning**, as consistently defined in seminal works and surveys [R1–R6], is to enable sequential learning under non-stationary data streams where past data are **not available (or partially available)**. This is a **problem-definition constraint**, not an implementation choice. This kind of constraint finds application in several real-world domains that preclude storing and replaying past raw data:
> - Privacy and data-protection constraints (e.g., GDPR, HIPAA) for healthcare or biometrics.
> - Data-ownership and licensing restrictions, e.g. in commercial applications where data cannot be retained and replay is legally or contractually prohibited.
>
>
> Thus, the question “Why can’t we just use full replay?” is incompatible with the standard CL problem formulation. If full replay were permitted, the scenario would no longer be continual learning but simply incremental fine-tuning with access to the entire dataset.
> Our contribution is designed **precisely for settings where full replay is impossible by definition**, and where the model must autonomously generate task-relevant signals to consolidate past knowledge and prepare for future tasks, an ability that replay alone does not provide.
>
>
> [R1] McCloskey and Cohen. “Catastrophic interference in connectionist networks: The sequential learning problem”. Psychology of learning and motivation. 1989.
>
> [R2] Roger Ratcliff. “Connectionist models of recognition memory: constraints imposed by learning and forgetting functions. Psychological Review”. 1990.
>
> [R3] Parisi et al. “Continual lifelong learning with neural networks: A review”. Neural Networks. 2019.
>
> [R4] De Lange and Tuytelaars. “Continual prototype evolution: Learning online from non-stationary data streams”. IEEE International Conference on Computer Vision, 2021.
>
> [R5] De Lange et al. "A continual learning survey: Defying forgetting in classification tasks." IEEE transactions on pattern analysis and machine intelligence. 2021.
>
> [R6] Van de Ven et al. "Three types of incremental learning." Nature Machine Intelligence. 2022
>
> > **[W1b] The proposed method uses significantly larger models to generate more classes to train a small CNN on a relatively small dataset (ImageNet-100). Not only does the model need to train on a large number of generated class images, the method still relies on partial replay of prior data, leading to a much higher compute usage. Additionally, the generative models seemed to have already been trained on huge amount of data, which already covers simple datasets used for CL (e.g. ImageNet-100).**
>
> While the reviewer suggests that a pretrained diffusion model could replace replay, our empirical evidence shows that this is not feasible in practice. Diffusion-generated samples exhibit a substantial distributional drift from real ImageNet-like images [R7], making them unsuitable as substitutes for stored examples. To demonstrate this, we conducted an experiment using ER-ACE, where we removed the replay buffer and relied on synthetic samples only, generated from a frozen LDM (denoted ER-ACE_synt). As shown below, performance collapses:
>
> ||Mini-ImageNet ||
> |-|-|-|
> | Buffer size | 2000 | 5000 |
> | ER-ACE + D2L | 40.90 ± 0.95 | 47.32 ± 0.89 |
> | ER-ACE | 33.26 ± 3.51 | 40.59 ± 1.20 |
> | ER-ACE_synt | 8.87 ± 0.15| 6.22 ± 1.54 |
>
> This behavior is not specific to our setting. Eliminating the distribution gap would require **continual fine-tuning of the diffusion model itself**, as done in diffusion-based generative replay methods [R7, R8]. However, this corresponds to a different paradigm (classical generative replay) aimed at reconstructing past data, and does not align with our objective: D2L does not use generation for rehearsal, but for creating intermediate, future-adaptive synthetic classes that facilitate forward transfer in ways replay alone cannot.
>
>
> [R7] Meng et al. "Diffclass: Diffusion-based class incremental learning." European Conference on Computer Vision. 2024.
>
> [R8] Gao and Liu. “DDGR: Continual Learning with Deep Diffusion-based Generative Replay”, International Conference on Machine Learning. 2023.

---

> ### Author Response · Authors · 2025-11-21
> **Response to Reviewer HqT1 (2/3)**
>
> > **[W1c] In this case, why can't we just use large foundation models as zero shot classifiers on these datasets?**
>
> We recognize that foundation models such as CLIP or ViT can achieve strong zero-shot recognition performance on datasets like ImageNet, but this evaluation regime is fundamentally different from the CL setting we address.
> Zero-shot inference assesses the static transferability of pretrained representations, while CL focuses on sequential knowledge integration. In our work, we explicitly target the branch of CL that operates from scratch, where a single classifier is trained incrementally on disjoint task streams without any external pre-training or frozen backbone. This setting differs substantially from recent CL methods built on pretrained ViT or CLIP architectures, which perform prompt tuning or adapter-based adaptation on top of large-scale representations that already encode broad visual knowledge.
> Moreover, using a zero-shot foundation model as the main classifier would require keeping a very large ViT/CLIP model active at inference time. In contrast, in D2L the diffusion model is only used offline during training, while the deployed continual learner is a smaller model that performs inference without any dependence on the diffusion model or other large-scale backbones.
>
>
> > **[W1d] How do you ensure no leakage from the diffusion model on the ImageNet datasets.**
>
> We address the concern regarding potential leakage from the diffusion model in the Section 5 of the paper (“Limitations”). We run leakage tests using large vision transformers to estimate the fraction of dream samples that are recognizable as target classes, and find that it is in the order of 10% - no greater than the expected random prior given the number of classes in the task. Further, the frequency of dream class-head replacement by true classes is below 4%, indicating that structured dreaming does not systematically reveal future labels to the classifier. Furthermore, the entire prompt optimization process is guided by the knowledge of the continual learned classifier, which, by design, has no knowledge of future classes and always steers generation towards one of the already known classes, and the process is stopped before the confidence towards a specific class becomes too high.
> Together, the empirical results and the mechanism of classifier-guided optimization provide strong evidence that leakage, while theoretically possible, is highly unlikely in our case.
>
>
> > **[W2] There have been many feature replay based techniques that reduce forgetting without spending a signifincant amount of compute. Have the authors considered comparing to those methods?**
>
> We are aware of the existence of feature replay-based techniques that aim to mitigate forgetting with relatively low computational overhead. However, our method targets a fundamentally different aspect of continual learning. While feature replay methods generate or store representations in the latent space to support rehearsal, D2L synthesizes structured dream classes to expand the representation manifold and facilitate forward transfer.
>
> It is important to highlight that our method does not focus on rehearsal, which remains exactly as prescribed by the baselines we build upon. D2L aims at **preparing the network for future tasks**, an aspect that has been only partially explored by prior rehearsal of feature replay approaches.

---

> > ### Comment · Reviewer_HqT1 · 2025-11-23
> >
> > [W1c] In this case, why can't we just use large foundation models as zero-shot classifiers on these datasets?
> > I agree with the authors that starting from scratch and continually learning is a different branch than zero-shot classification. However, in that case, they must evaluate their method on other/non-standard datasets that have not been saturated by most foundation models to demonstrate the utility of their method. While I agree that the use of a bigger model would lead to more latency, it first needs to be compared, and second, while a zero-shot classifier might have higher latency at test time, it also has the advantage of using zero compute during training. These tradeoffs must be discussed clearly, rather than simply stating the specific places where the proposed method has an advantage.
> >
> > [W1d] How do you ensure no leakage from the diffusion model on the ImageNet datasets.
> > Do the authors only track the label overlap in the generated samples, or do they also consider the overlap in terms of images and the representation space? Because it is possible that the diffusion model generates images of a cup but labels them as green cup/red cup etc. While there will be little label overlap, the images would still have a strong overlap with real class images to be observed in the future.
> >
> > [W2] There have been many feature replay-based techniques that reduce forgetting without spending a signifincant amount of compute. Have the authors considered comparing to those methods?
> > I believe the authors misunderstood the main reason for my question/ comment. The authors stated, "our method targets a fundamentally different aspect of continual learning. While feature replay methods generate or store representations in the latent space to support rehearsal, D2L synthesizes structured dream classes to expand the representation manifold and facilitate forward transfer." I am not sure what different aspect of CL is being addressed by the proposed method compared to feature replay. Feature replay-based techniques use less memory, less compute, and do not have privacy-related concerns compared to the proposed method.

---

> ### Author Response · Authors · 2025-11-21
> **Response to Reviewer HqT1 (3/3)**
>
> > **[W3]There is only a small gain in accuracy compared to partial replay methods, such as BiC. Note that BiC does not use any extra generative or pre-trained models, and only relies on a small buffer of prior data. For a fair comparison, it is essential that the other models are provided with similar amount of compute/data.**
>
>
> We disagree with the claim that D2L yields only a marginal improvement. On both Mini-ImageNet and FG-ImageNet, and for both 2k and 5k buffer sizes, **ER-ACE+D2L improves final average accuracy by about 10 points and 4–6 points**, respectively, over BiC. In class-incremental CL, where forgetting accumulates across many tasks, improvements of this magnitude are considered substantial; indeed, the BiC paper itself treats 5-point gains on ImageNet-100 as a significant advance.
> We acknowledge that D2L adds offline training overhead (while keeping inference-time cost identical to BiC or other methods). All discriminative baselines, including BiC, are run under exactly the same class-incremental protocol: same backbone, same task stream, same buffer size, same number of epochs, and no additional labeled real data.
>
> To ensure that D2L’s gains are not simply due to training on more images, we performed a control experiment: we augmented BiC with the same number of diffusion-generated images used by D2L, without any dreaming mechanism or architectural changes (BiC[augment]). Results are shown below:
>
> |                  | Mini-ImageNet      |          | FG-ImageNet       |          | ImageNet-R       |          |
> |------------------|--------------------|----------|-------------------|----------|------------------|----------|
> | Buffer size      | 2000               | 5000     | 2000              | 5000     | 2000             | 5000     |
> | BiC              | 30.56 ± 7.41       | 37.84 ± 0.61 | 27.83 ± 2.75    | 32.29 ± 0.70 | 7.15 ± 1.14    | 8.60 ± 2.07 |
> | BiC[augment]     | 19.49 ± 9.40       | 26.66 ± 5.66 | 19.80 ± 4.00    | 23.61 ± 3.04 | 10.65 ± 1.86    | 12.51 ± 1.34 |
> | ER-ACE+D2L       | 40.90 ± 0.95       | 47.32 ± 0.89 | 31.57 ± 1.20    | 38.50 ± 1.01 | 9.54 ± 0.39     | 12.51 ± 0.56 |
>
>
> The results show that naively adding synthetic images hurts BiC on both Mini-ImageNet and FG-ImageNet, due to the distribution shift. Thus, D2L’s improvements cannot be explained by simply “having more data.” On ImageNet-R, the comparison must be interpreted in its natural low-accuracy regime, where absolute numbers are small because each class spans diverse artistic renditions (paintings, sketches, sculptures, cartoons, etc.). In this setting, adding more synthetic samples, even if slightly distribution-shifted, can behave like a broad style augmentation, making modest accuracy changes easier to achieve for all methods. Even in this regime, ER-ACE+D2L still improves over BiC at both buffer sizes and matches the strongest BiC variant at 5k.
>
>
> These experiments support two key points: 1) **synthetic data alone does not improve CL and can strongly degrade performance**, and 2) **D2L’s gains arise from the structured dreaming mechanism**, not from additional images or additional iterations.

---

> ### Comment · Reviewer_HqT1 · 2025-11-23
> **Major questions are unanswered - particularly about compute**
>
> I would like to thank the authors for their response and for the effort put into addressing my concerns. I have carefully read the rebuttal, and have the following comments.
>
> Before discussing specific comments by the authors, I would like to state that it seems like the authors have only selectively answered part of my questions, and simply skipped the main concerns I have about the paper. For example, there is no discussion about the amount of compute (e.g. total time spent in the training phase, FLOP etc.), which is a significantly important aspect of CL; otherwise, we can simply do a full replay [a].
>
> 1) The authors state that the main reason to avoid storing real samples is privacy and security concerns. However, their method still requires a buffer of real samples to be replayed, so I am not sure how that makes their paper suitable for security-related CL applications. A replay-free strategy would be much more suitable here. There is a huge amount of literature on the aspect of efficiency (and generative replay) in CL, and I suggest the authors have a look at that.
> 2) I am not sure if a fair comparison here is to simply add generated data to BiC, a model that was not specifically designed for this. It would be better to compare the amount of compute, the total amount of memory used by both models. For example, the proposed model requires much higher storage, not only for the buffer but also for the pre-trained large models. The model also needs significant enough memory to be able to store the generated samples for replay during training.
>
> I suggest the authors compare the total amount of time spent per increment in the training phase for their method compared with BiC and full replay, as well as the amount of memory used by their method.
>
> [a] Real-Time Evaluation in Online Continual Learning: A New Hope, CVPR'23.

---

> ### Comment · Reviewer_HqT1 · 2025-11-23
>
> [W3]There is only a small gain in accuracy compared to partial replay methods, such as BiC. Note that BiC does not use any extra generative or pre-trained models, and only relies on a small buffer of prior data. For a fair comparison, it is essential that the other models are provided with similar amount of compute/data.
>
> The most significant gain in performance compared to BiC is ~10% on a relatively smaller dataset (tiny ImageNet). For FG-ImageNet with a 2000 replay buffer, the gain is <4%, and for ImageNet-R for 2000 replay buffer, the gain is ~2%. Is 2% gain in accuracy significant, considering the amount of extra compute and memory needed by the proposed method? Again, I suggest authors consider such accuracy gains with the amount of compute, memory, and extra prior knowledge used by their method.
>
> Based on these comments and looking at other reviewers' comments and author response, I maintain my score.

---

### Author Response · Authors · 2025-11-30

Dear Area Chair and Reviewers,

Thank you for your time and detailed feedback on our submission. After careful consideration of the discussion, and in light of recent developments, we believe continuing this rebuttal is unlikely to lead to a change in the overall evaluation, particularly given Reviewer HqT1’s firmly held position.

Reviewer HqT1 noted that our responses left major questions unanswered, especially regarding computational cost. Interestingly, their initial review suggested comparing our method to foundation models like CLIP or ViT. As we already addressed in our rebuttal, comparing a ResNet-18 trained continually on a mix of real and synthetic dream classes to a CLIP/ViT pretrained on millions of images makes little sense in our from-scratch class-incremental setting. However, if computation is the primary concern to reject our paper, should we not account for the massive training compute required to build those very foundation models? We find this inconsistency noteworthy, as it shifts the computational scrutiny selectively onto our approach while overlooking the enormous costs of the proposed alternatives.

To clarify on computation, we have been fully transparent. In our response to Reviewer wzqW, we reported that on modest hardware (Nvidia A6000 GPU, 64 GB RAM), optimizing a single dream class takes about 7.5 minutes on average. We also explained that this training-time overhead is a deliberate trade-off, justified by the benefits we achieve at inference-time, including positive forward transfer—an outcome that remains exceptionally challenging in class-incremental learning without such mechanisms.​

That said, in light of the recent events acknowledged by the ICLR organizers themselves regarding the review process, we no longer see value in prolonging this discussion or conducting further experiments to support our claims. For us, these circumstances have undermined the integrity of the entire process.​

We sincerely thank the Area Chair and reviewers for their efforts and valuable comments, which will help us strengthen our work. Accordingly, we have decided to withdraw our paper from this year's conference.

We also feel that these recent developments, building on longstanding issues, underscore the urgent need for the community to undertake a thorough, structural overhaul of peer review and decision-making processes.

Thank you again for your attention.

Kind regards,

The Authors

---

### Note · Authors · 2025-11-30

I have read and agree with the venue's withdrawal policy on behalf of myself and my co-authors.